# Integrated crop-livestock farming systems influence the incidence of *Salmonella*, *Listeria monocytogenes*, Shiga toxin-producing *Escherichia coli*, and indicator bacteria on fresh produce

Brian Goodwyn,[1] Patricia Millner,[2] Anuradha Jeewantha Punchihewage-Don,[1] Melinda Schwarz,[1] John Bowers,[3] Joseph Haymaker,[4] Fawzy Hashem,[1] Chyer Kim,[5] Debabrata Biswas,[6] Salina Parveen[1]

**ABSTRACT** Concerns remain about foodborne pathogen contamination risks to integrated crop-livestock farm (ICLF) fresh produce due to farm animal proximity to crop fields and use of biological soil amendments of animal origin (BSAAO). This study aimed to evaluate the extent of microbial contamination in Maryland's Eastern Shore ICLFs and compare results with those from corresponding samples from crop-only farms (COFs), farmers' markets, and supermarkets. Altogether, 1,782 soil, animal reservoir, water, and produce samples from ICLFs, COFs, farmers' markets, and supermarkets were analyzed following standard methods. Total aerobic bacterial counts and generic *Escherichia coli* were enumerated using petrifilms, whereas *Salmonella, Listeria monocytogenes* (*L. monocytogenes*), and Shiga toxin-producing *E. coli* (STEC) and virulence factors (VFs) were detected by culturing and PCR confirmation. ICLF soil health parameters were evaluated pre- and post-BSAAO incorporation. Animal pen samples and BSAAOs, which generally improved ICLF soil health parameters, harbored more pathogens and generic *E. coli*. Samples of ICLF produce (0.39%, 1.95%, and 13.62%) and soils (2.04%, 2.72%, and 20.86%) had higher *Salmonella*, *L. monocytogenes*, and STEC/VF-gene prevalence than COF produce (0.00%, 0.00%, and 5.33%) and soils (1.33%, 0.00%, and 20.00%), respectively. Pathogen contamination was relatively low in retail environments where *Salmonella* and *L. monocytogenes* were each isolated from one farmers' market produce, whereas STEC/VF-genes were found in one supermarket produce. Overall, the most frequent *Salmonella* serovars were Bareilly and Newport, whereas the highest detected STEC serovars and VF-genes were O103 and *stx*2. Produce contamination in ICLFs and farmers' markets was greater than that of traditional counterparts, indicating the importance of continued research/education regarding produce safety for producers and consumers.

**IMPORTANCE** Shifts in consumer demand have led to notable increases in integrated crop-livestock farms (ICLFs), which are major contributors to popular farmers' markets. However, production practices in these settings, like manure-based soil amendments and the lack of food safety regulation enforcement, have raised concerns regarding potential foodborne illness cases and outbreaks. This study provides valuable information on the prevalence and comparison of three major foodborne bacterial pathogens and indicator microorganisms in ICLF, crop-only farm (COF), farmers' market, and supermarket environments. *Salmonella*, *Listeria monocytogenes*, and Shiga toxin-producing *E. coli* were detected more frequently in ICLF and farmers' market samples compared with COF and supermarket samples, respectively. However, at least one pathogen was detected in each setting. Findings from this study highlight that, regardless of studied

**Peer Reviewer** Stanley Chen, Queensland Alliance for Agriculture and Food Innovation, Brisbane, Australia

Address correspondence to Salina Parveen, sparveen@umes.edu.

The authors declare no conflict of interest.

See the funding table on p. 21.

*[This article was published on 27 October 2025 with incomplete funding details and typographical errors in the text. The funding details and text were corrected in the current version, posted on 3 November 2025.]*

settings, the contamination risks associated with fresh produce production on the Maryland Eastern Shore will contribute to developing food safety standards in various produce production environments.

**KEYWORDS** *Salmonella*, *Listeria monocytogenes*, STEC, biological soil amendments, fresh produce, farmers markets, food safety

Consumer demand for local, organic produce increased from $11 billion in 2004 to $52 billion in 2021 (1–3). Unlike conventional counterparts, organic regulations prohibit synthetic fertilizers, pesticides, antibiotics, etc. Consumers seek local produce to support local farm sustainability, freshness, and environmental quality (4, 5). Consequently, U.S. farmers' markets have increased significantly from 1,755 sites in 1994 to 8,755 in 2019, and a positive relationship between the number of farmers' markets and reported foodborne illnesses has been reported (4–7). Research investigating microbial hazards of farmers' markets reported *Campylobacter*, generic *Escherichia coli* (*E. coli*), *Listeria monocytogenes* (*L. monocytogenes*), and *Salmonella* on produce in multiple states (2, 5, 6, 8, 9). Researchers found farmers' market produce is more contaminated with *Campylobacter*, total coliforms, generic *E. coli, L. monocytogenes*, and *Salmonella* than traditional retail-store samples (2, 8, 9). Higher farmers' market produce contamination might be attributed to a lack of training and implementation of Good Agricultural Practices (GAP) and Good Handling Practices (GHP) (5, 6).

Integrated crop-livestock farm (ICLF) popularity has increased because they tend to be organic and contribute significantly to farmers' markets (2, 10–12). These sustainable farms recycle on-farm resources naturally by growing crops and raising animals within the same farming system, where the two components support each other's growth. Crops are used as animal feed, whereas animals graze fields and generate manure for biological soil amendments of animal origin (BSAAO) that improve agricultural soil nutrient content, structure, and other health parameters that influence crop yields. ICLF producers can also financially benefit from consumer demand and increased potential net farm profit with reduced input costs (feed, fertilizer, and pesticide) partnered with multiple products (crops, meats, dairy, and eggs) (2, 11–13).

Despite perceptions and benefits, ICLF fruits and vegetables have a higher chance of causing foodborne illness when grown near farm animals, typically harboring foodborne pathogens (2, 11, 12). Produce contamination is often traced back to animal reservoirs, and studies have isolated pathogens (e.g., *Salmonella*, *Campylobacter*, or Shiga toxin-producing *E. coli* [STEC]) from animal pens, feces, feed, forage, and hide (12, 14–17). On ICLFs, animals and produce may share pre- and post-harvest facilities/tools, thus increasing cross-contamination risks without proper handling and sanitization practices (2, 11). Raw or improperly stored BSAAO can harbor and introduce pathogens that can survive in soils for several months (18). Pathogens in animal reservoirs and soils might colonize seeds and/or transfer to food crops through contact, splash, or dust (2, 10, 15, 17). Studies have detected various foodborne pathogens in ICLF-BSAAO, soils, and produce and have found *Salmonella* and *Campylobacter* to be more prevalent in ICLFs (soil, water, and produce) than monocultural farms (2, 10, 12, 17, 19).

To reduce ICLF contamination risks, government agencies have set standards regarding BSAAO, water quality, staff hygiene, equipment, etc. The U.S. Food and Drug Administration Food Safety Modernization Act Produce Safety Rule provides science-based safety measures for the safe growing, harvesting, packing, and holding of produce for human consumption (20–22). Although the Produce Safety Rule increases food safety, the standards are enforced based on business size; hence, small-scale ICLFs and their products sold at farmers' markets are exempt (2, 11). Increased demand and lack of regulations surrounding these products increase the risks of foodborne illnesses. Furthermore, the U.S. Department of Agriculture National Organic Program mandates a 90- or 120-day withholding period between the application of untreated BSAAO and the harvest of crops whose edible portions indirectly or directly contact soils, respectively

(10, 17, 23). This withholding period is based on farm production cycles rather than experimental evidence, and research has shown pathogens to persist after the withholding period (10, 14, 24–26).

*Salmonella, L. monocytogenes*, and STEC are three common pathogens responsible for produce-related multistate outbreaks. From 1998 to 2017, there was a positive relationship between produce consumption and the number of produce-associated outbreaks, hospitalizations, and deaths linked to these pathogens (20, 21). Produce contamination can occur from soil/soil-splash, BSAAO, tools, aerosols, etc., and be accentuated by post-harvest conditions. Mitigation strategies are needed. The current study assessed microbiological contamination of ICLF animal pen, manure/compost (BSAAO), produce, soil, and water samples. To assess sample hygiene, total aerobic bacteria (APC), generic *E. coli,* and total coliform counts were enumerated. *Salmonella, L. monocytogenes,* and STEC serovars and associated virulence-factor genes (STEC/VF-genes), which contribute to pathogenicity, were tested to evaluate food safety risks in each environment. Samples from ICLFs were compared with corresponding samples collected from crop-only farms (COFs), and farmers' market produce was compared with supermarket produce to evaluate contamination differences and risks in unique ICLF and farmers' market environments. In addition, ICLF soil health parameters were evaluated pre- and post-BSAAO incorporation.

## MATERIALS AND METHODS

### Sampling location selection criteria, description of farms, and study population

All samples were collected from the Maryland Eastern Shore fresh produce farm (ICLF and COF) and retail (farmers' market and supermarket) settings. Farms chosen met the following criteria: (i) be organic, (ii) apply BSAAOs to crop soils, whereas ICLFs and COFs must (iii) grow fresh produce, (iv) generate/sell products locally, and (v) provide on-farm samples and management information. Recruits were personally contacted by phone, email, or in-person visit; three ICLFs and two COFs provided on-farm samples and farm descriptions that assessed management practices (irrigation, BSAAOs, presence/type of animals, etc.). Selected farms met the previously listed criteria and agreed to participate, whereas farms that were not selected either did not respond or did not have the needed samples available. To represent retail environments, produce samples, which were of the same type as produce collected on the farms, were obtained from three farmers' markets and supermarkets. All farmers' markets were listed in the Maryland Farmers' Market Directory. The UMES Extension Program facilitated the recruitment of ICLF, COF, and farmers' market locations based on existing relationships with farmers and vendors. In return for their contributions, farmers and vendors were provided soil fertility results of their crop soils before and after BSAAO application or were paid at their request. They also had access to the resources of UMES' Extension and Outreach Program for assistance, and participants received all results with individualized advice at the conclusion of the study.

Collected samples varied in type and number because available samples (animal reservoir, produce, soil, etc.) differed across all locations. For anonymity, locations were identified as ICLF-A, ICLF-B, ICLF-C, COF-D, COF-E, FM-1, FM-2, FM-3, SM-1, SM-2, and SM-3. Three spatially separated ICLFs were organic (not USDA-NOP certified) and provided animal pen, BSAAO, produce, soil, and water samples. Two COFs were conventional farms that supplied produce, soil, and water samples. The farmers' market and supermarket locations provided produce. All locations grew and sold fresh produce: see Tables 1 to 3 for detailed descriptions of sampling locations, numbers, and types.

Samples were collected from March 2020 through December 2021 for ICLFs, June 2021 through November 2021 for farmers' markets, and September 2021 through December 2021 for COFs and supermarkets. The COVID pandemic constrained farmer and sample availability and enrollment times.

**TABLE 1** Collected samples for microbiological analysis across different sources and categories[a]

| Source | Sample category | Sample types | Sample numbers |
|---|---|---|---|
| ICLF (n = 3) | Animal Pen | Chicken coop, cow pen, pig pen | 124 |
| | Manure/compost | Chicken manure, cow/pig manure mix, mushroom compost, vegetable material/chicken manure mix | 119 |
| | Produce | basil, bell pepper, chili pepper, collard greens, greens mix, jalapeno pepper, kale, mustard greens, okra, romaine, squash, Swiss chard, tomato | 257 |
| | Soil | Soil | 441 |
| | Water | Well water | 18 |
| COF (n = 2) | Produce | Bell pepper, collard greens, kale, squash, tomato | 150 |
| | Soil | Soil | 150 |
| | Water | Well water | 20 |
| Farmers' Market (n = 3) | Produce | Bell pepper, kale, okra, romaine, squash, tomato | 230 |
| Supermarket (n = 3) | Produce | Bell pepper, kale, romaine, squash, tomato | 273 |
| Totals | | | 1,782 |

[a]Selected types of produce were chosen because they are eaten raw or grown in close contact with the soil.

## Sample collection

Nitrile gloves, soil cores/scoops, and other sampling equipment were cleaned and sanitized (70% ethanol) before and after each sampling. In total, 1,782 samples were collected from March 2020 to December 2021 (Table 1). Five sample types were collected: animal pen, BSAAO, produce, soil, and water. Animal pen and BSAAO samples were collected monthly from their designated locations on ICLFs. Soil samples were collected from the same selected plots in ICLF and COF fields monthly; water samples were collected in summer and fall from well irrigation sources. When available, produce samples were collected from either corresponding soil plots on ICLFs and COFs or from farmers' markets and supermarkets. All produce selected appeared ripe and ready to eat, without visible damage/spoilage, or contamination (bugs, pests, and feces), or processing. ICLF and COF produce samples were evaluated directly from soils; no soil remained on farmers' market and supermarket produce displayed by retailers. See Tables 1 to 3 for details.

All samples were collected and processed aseptically using standard methods described previously (10, 17, 27, 28). Animal pen, manure/compost, and soil samples were composited in replicates of 3 or 5 into 52 oz Whirl-Pak bags (NASCO Whirl-Pak, Madison, WI) using a stainless-steel probe/scoop. Animal pen samples consisted of soils collected at a depth of ~7 cm from high-trafficked areas (water/feed) of animal enclosures where manures were collected, stored, and aged for an ICLF's manure amendment. BSAAOs were collected from several depths and locations from ICLF stockpiles. Soils were collected at ~7–15 cm from the soil surface. On ICLFs, additional control soil samples were collected from a designated area where BSAAOs were never applied. Irrigation water samples were collected into sterilized Nalgene (Thermo Scientific, Waltham, MA) 1 L bottles. Spigots were sterilized (70% ethanol) and flushed for 2 min before collection. ICLF and COF produce samples were collected from soils into sterile Whirl-Pak bags (up to n = 11/plot) during each sampling, depending on availability. Farmers' market and supermarket produce samples were purchased in duplicate from each vendor or supplier at farmers' markets and supermarkets. Harvested samples were unwashed, whereas farmers' market and supermarket samples may have been washed at the source. All samples collected from their respective locations were placed on ice in insulated coolers and transported back to the Food Microbiology and Safety Laboratory at the University of Maryland Eastern Shore and processed within 24–48 h.

## Sample processing, enrichment, and bacterial isolation

All samples were enumerated for total APC, generic *E. coli*, and total coliform levels and evaluated for the presence of *Salmonella*, *L. monocytogenes*, and STEC/VF-genes.

**TABLE 2** Farm description[a]

|  | ICLF-A | ICLF-B | ICLF-C |
|---|---|---|---|
| Soil type | Sandy loam | Loam | Silt loam |
| Soil series | Mullica-Berryland complex | Fallsington | Nassawango |
| Slope | 0%–2% | 0%–2% | 0%–2% |
| Irrigation | Yes. Well water | No. | Yes. Well water |
| Manure or compost type | Mushroom compost | Cow and Pig manure mix | Homemade. Chicken manure, Plant leftovers, grass coffee filters, eggs, vegetable leftovers, wood ash |
| Age of manure or compost | N/A. Purchased from a third party | 1 year | Chicken manure (4 years) The rest (2–3 years) |
| How much manure applied (approximate, dry t ha-1) | 24 | 0.6 | 13 |
| Manure incorporation | Mechanically tilled | Mechanically tilled | Raked in |
| Cover crop before soil application | None | None | Lemon balm |
| Soil tilled/cultivated before application? | Tilled, application applied, and then tilled in | Tilled, application applied, and then tilled in | Raked, application applied, and then raked in |
| Duration of manure/compost peak heat periods | N/A, buy from seller. Uncovered. | N/A, aged manure. | N/A, aged manure |
| Storage method | Pile, uncovered | Two piles. Covered (Cow) and uncovered (pig) | Stored in buckets with closed lid |
| Certified organic | No | No | No |

[a]N/A, not available.

Total APC and coliforms were measured to evaluate sanitary conditions associated with samples, whereas generic *E. coli* indicated fecal contamination. Pathogen prevalence was assessed to evaluate food safety risks in the different farm and retail environments. Sample processing methods were derived from previous studies (5, 10, 14, 29) and were either stomached (soil, animal reservoir, and leafy greens, herbs) or hand massaged vigorously (bell peppers, tomatoes, squashes, chili peppers, jalapeño peppers, and okras), depending on the suitability of the sample for the processing method.

## Petrifilms—enumeration of APC, generic E. coli, and total coliform levels

For stomaching, 25 g of samples were suspended with 225 mL 0.1% sterile peptone water in a Whirl-Pak bag, stomached for 2 min at 230 rpm, and used to prepare 10-fold serial dilutions. For massaged samples, produce was weighed into sterile Whirl-Pak

**TABLE 3** Animals and produce grown on three Maryland eastern shore integrated crop-livestock farms (ICLF)

|  | ICLF-A | ICLF-B | ICLF-C |
|---|---|---|---|
| Farm animals | Goats, pigs, chickens, turkeys, ducks, cats | Goats, pigs, cows, chickens, turkeys, ducks, cats | Chickens, cats |
| Animal coop/pens sampled | Chicken | Pig, cow | Chicken |
| Crops grown | Greens mix, Swiss chard, kale, romaine lettuce, bell pepper, tomato, squash, collard greens, corn, apples, strawberries, blackberries, other lettuces, onion | Bell pepper, jalapeno pepper, okra, chili pepper, squash, tomato | Swiss chard, mustard greens, basil |
| Crops sampled | Greens mix, Swiss chard, kale, romaine lettuce, bell pepper, tomato, squash, collard greens | Bell pepper, jalapeno pepper, okra, chili pepper, squash, tomato | Swiss chard, mustard greens, basil |

bags with 0.1% sterile peptone water added in equal mass amounts, and then hand-massaged for 2 min. Ten-fold serial dilutions were then prepared; all samples were plated for enumeration on 3M Petrifilm Aerobic Count Plate and *E. coli*/Coliform Count plate Petrifilms (3M St. Paul, MN), then incubated and counted following manufacturer's instructions.

## Water samples

For irrigation water samples, generic *E. coli* and total coliform levels were assessed using Colilert* Quanti-Tray/2000 MPN (IDEXX Laboratories, Westbrook, ME) with a detection limit of one organism per 100 mL (30). Samples were processed and interpreted according to the manufacturer's instructions. None of the water samples were positive for generic *E. coli*; hence, irrigation water samples were not assessed for pathogen prevalence.

## Pathogen isolation

For stomached samples, 30 g of the sample and 120 mL of primary enrichment broth were added into 24 oz sterile Whirl-Pak bags. As for non-stomached samples, 30 mL of primary sample suspension (0.1% peptone water) used for Petrifilm analyses was added to Whirl-Pak bags with 120 mL primary enrichment broth (2, 10, 12, 31).

## Salmonella spp.

*Salmonella* enrichment and isolation methods were used with some modifications from previously published literature (2, 10, 12). Samples were enriched with buffered peptone water (Becton, Dickinson and Company, Sparks, MD) and incubated at 37°C for 24 h. Thereafter, 100 µL of sample was added to 10 mL Rappaport Vassiliadis broth (Becton, Dickinson and Company, Sparks, MD) and incubated for 24 h at 42°C; the samples were streak-plated onto Xylose Lysine Tergitol-4 agar (Becton, Dickinson and Company, Sparks, MD) and incubated at 37°C for 48 h. Presumptive black colonies were picked for confirmation

## L. monocytogenes

*L. monocytogenes* enrichments and isolations followed modified methods from Ramos et al. (10). Using *Listeria* Enrichment broth (Becton, Dickinson and Company, Sparks, MD), samples were enriched, stomached for 2 min, and incubated at 37°C for 24 h. Following incubation, 200 µL samples were pipetted into 5 mL of Fraser Broth (Oxoid Ltd, Hampshire, United Kingdom) and then incubated for 48 h at 37°C. Positive black tubes were streaked onto Brilliance *Listeria* Agar (Oxoid Ltd, Hampshire, United Kingdom) and incubated at 37°C for 48 h. Presumptive blue/green colonies with a halo were picked for confirmation.

## Shiga toxin-producing E. coli and virulence-factor genes (STEC/VF-genes)

The STEC/VF-gene enrichment and isolation methods were derived from Aditya et al. (31) with a few modifications. Samples were enriched with Luria-Bertani broth (Becton, Dickinson and Company, Sparks, MD) supplemented with 5% sheep blood (Colorado Serum Co, Denver, CO), stomached for 2 min, and incubated for 24 h at 37°C. Following incubation, the samples were streak-plated onto MacConkey with Sorbitol Agar (Becton, Dickinson and Company, Sparks, MD) and incubated for 24 h at 35°C, with pink colonies presumed STEC. Three presumptive isolates were spread plated onto Tryptic Soy Agar (Becton, Dickinson and Company, Sparks, MD) for all pathogens, and colonies were frozen at −80°C in cryovials containing tryptic soy broth (Becton, Dickinson and Company, Sparks, MD) with 24% glycerol (Fisher, Fair Lawn, NJ).

## Confirmation of presumptive pathogens

### *Salmonella spp.*

Presumptive *Salmonella* was confirmed through the BAX System PCR (Dupont Qualicon, Wilmington, DE) assay for *Salmonella*. From freezer vials, 10 µL loops were used to inoculate tryptic soy broth tubes with freezer vial cultures, which were incubated overnight at 37℃ for confirmation through BAX. This procedure was conducted according to the manufacturer's instructions. Briefly, 5 µL of enriched sample was added to 200 µL of lysis reagent in cluster tubes that were heated at 37℃ for 20 min, 95℃ for 10 min, and cooled in a cooling block (4℃) for 5 min. PCR tablets were hydrated with 50 µL of lysate, loaded in the BAX System, run, and interpreted based on methods detailed in the user guide.

### *Salmonella serotyping*

BAX PCR-confirmed *Salmonella* isolates were serotyped at the United States Department of Agriculture National Veterinary Services Laboratories (NVSL, AMES, IA) in accordance with the standardized World Health Organization guidelines (32).

### *L. monocytogenes*

The *L. monocytogenes* presumptive isolates were confirmed by a polymerase chain reaction (PCR) that amplified the *hly*A (388 bp fragment) gene based on methods described by Aznar and Alarcon (33). Sample DNA was extracted using Instagene Matrix (Bio-Rad, Hercules, CA) according to the manufacturer's instructions. The primers for this reaction were F, 5′-GAATGTAAACTTCGGCG-CAATCAG-3′ and R, 5′-GCCGTCGATGATTTGAA CTTCATC-3. The PCR reaction had a final volume of 25 µL that contained 12.5 µL GoTaq Green Mastermix (Promega, Madison, WI), 8.5 µL PCR Water (Invitrogen, Waltham, MA), 0.5 µL of each primer, and 3 µL of sample DNA. The thermocycler (Applied Biosystems, Waltham, MA) program was set for an initial denaturation at 95℃ for 5 min, followed by 30 cycles of denaturation at 95℃ for 45 s each, annealing at 56℃ for 45 s, extension at 72℃ for 60 s, and final extension at 72℃ for 7 min. PCR products were held at 4℃ until they were separated by gel electrophoresis in a 1% agarose gel. The gels were stained with ethidium bromide and viewed using a BioSpectrum 310 Imaging System (UVP, LLC, Upland, CA).

### *Shiga toxin-producing E. coli and virulence-factor genes*

Presumptive STEC isolates were confirmed and identified through an 11-plex PCR protocol targeting seven major STEC serogroups and four virulence-factor genes. Primer sequences and PCR methods were derived from protocols described in previous literature (34, 35). This PCR assay screened for sequences unique to STEC serovars O45, O103, O121, O145, O26, O111, and O157:H7, along with sequences unique to VF-genes *stx1*, *stx2*, *eae*, and *ehx*A. These selected serogroups and VF-genes were chosen because they have been frequently associated with and contribute to the development of STEC outbreaks and severe cases such as hemolytic uremic syndrome (HUS) and hemorrhagic colitis (36–38). Primer (Integrated DNA Technologies, Coralville, IA) sequences, amplicon size (base pair), and target genes are described in Table 4. Presumptive freezer vial isolates were plated onto tryptic soy agar for enrichment, incubated at 37℃ for 24 h, and DNA was extracted from cultures using InstaGene Matrix (Bio-Rad Laboratories, Hercules, CA) according to the manufacturer's instructions. The PCR reaction had a final volume of 20 µL, containing 12.4 µL PCR water, 4 µL 5× My*Taq* Reaction buffer (Meridian Bioscience, Cincinnati, OH), 1.2 µL concentration of $MgCl_2$ (Meridian Bioscience, Cincinnati, OH), 0.4 µL My*Taq* HS DNA polymerase (Meridian Bioscience, Cincinnati, OH), 1 µL of primer mix (concentration of 0.42 µM for O111 and 0.21 µM for the other primers), and 1 µL of sample DNA. Thermocycler (Applied Biosystems, Waltham, MA) parameters were set for initial denaturation for 1 min at 95℃, followed by 35 cycles of denaturation, annealing, and extension for 15 s at 95℃, 15 s at 62℃, and 90 s at 72℃, respectively.

TABLE 4   11-plex PCR primers used for the confirmation of Shiga toxin-producing *E. coli* serovars and virulence-factor genes

| Primer designation | Sequence ("→") | Amplicon size (base pairs) | Target gene/ VF |
|---|---|---|---|
| O45-F | GGGCTGTCCAGACAGTTCAT | 890 | *E. coli* O45 (wzxO45) |
| O45-R | TGTACTGCACCAATGCACCT | | |
| O103-F | GCAGAAAATCAAGGTGATTACG | 740 | *E. coli* O103 (wzxO103) |
| O103-R | GGTTAAAGCCATGCTCAACG | | |
| stx1-F | TGTCGCATAGTGGAACCTCA | 655 | stx1 |
| stx1-R | TGCGCACTGAGAAGAAGAGA | | |
| O121-F | TCAGCAGAGTGGAACTAATTTTGT | 587 | *E. coli* O121 (wbqEO121 |
| O121-R | TGAGCACTAGATGAAAAGTATGGCT | | wbqFO121) |
| O145-F | TCAAGTGTTGGATTAAGAGGGATT | 523 | *E. coli* O145 (wzxO145) |
| O145-R | CACTCGCGGACACAGTACC | | |
| stx2-F | CCATGACAACGGACAGCAGTT | 477 | stx2 |
| stx2-R | TGTCGCCAGTTATCTGACATTC | | |
| O26-F | AGGGTGCGAATGCCATATT | 417 | *E. coli* O26 (wzxO26) |
| O26-R | GACATAATGACATACCACGAGCA | | |
| eae-F | CATTATGGAACGGCAGAGGT | 375 | eae |
| eae-R | ACGGATATCGAAGCCATTTG | | |
| rfbE-F | CAGGTGAAGGTGGAATGGTTGTC | 296 | *E. coli* O157 (rfbO157) |
| rfbE-R | TTAGAATTGAGACCATCCAATAAG | | |
| O111-F | TGCATCTTCATTATCACACCAC | 230 | *E. coli* O111 (wzxO111) |
| O111-R | ACCGCAAATGCGATAATAACA | | |
| ehxA-F | GCGAGCTAAGCAGCTTGAAT | 199 | ehxA |
| ehxA-R | CTGGAGGCTGCACTAACTCC | | |

PCR products were held at 4°C until electrophoresis. PCR products were electrophoresed in 1× LB buffer (Faster Better Media, Hunt Valley, MD) on a 1% agarose gel containing 1× GelRed Nucleic Acid Gel Stain (Biotium, Fremont, CA). Gels were viewed using UVP GelStudio Plus Touch (Analytik Jena US, Upland, CA) and Vision Works Capture and Analysis Software (Analytik Jena US, Upland, CA).

## Soil health testing

Additional composite control, pre-, and post-BSAAO incorporation soil samples were collected in the same manner and same plots as microbiologically analyzed soils from ICLFs into sterile Whirl-Pak bags. Soil samples were analyzed by AgroLabs (Harrington, DE) for a range of soil properties, including pH in water (1:1), buffer pH, cation exchange capacity, soil organic matter, and Mehlich-3 extractable phosphorus, potassium, magnesium, calcium, and sodium (39–43).

## Statistical analysis

Prevalence of pathogens and generic *E. coli*, and colony counts of APC and total coliforms were analyzed unadjusted for potential differences between collection sites by pooling sample data across farms (ICLFs, COFs) and retail environments (farmers' markets, supermarkets). Analyses were applied to identify the statistical significance of differences in the abundance of APC and total coliforms and the prevalence of pathogens (*Listeria*, *Salmonella*, STEC, and generic *E. coli*) in samples collected from different sources or locations (ICLF, COF, farmers' markets, supermarkets, and reservoirs). Comparisons of interest varied depending on the measurement and sample type (produce, BSAAO, animal pen, and soil). These analyses were also applied to fall-only subsets of data, given that the COVID pandemic and farm availability limited seasonally balanced sample collection, and COF samples were only collected in the fall. Statistical analysis of abundance of APC and total coliforms was performed by applying an analysis

of variance (ANOVA) to log-transformed counts, after substituting a value of 10 (the limit of detection) for counts of 0. ANOVAs were applied separately for each sample type, with statistical significance of differences in mean log abundance determined pairwise by post-hoc *t* tests. There were three comparisons of interest for produce samples (ICLF vs COF, farmers' market vs supermarket, and preharvest vs postharvest) and one each for reservoir (manure/compost vs animal pen) and soil (ICLF vs COF) samples. Statistical analysis of pathogen/generic *E. coli* prevalence was performed by applying Fisher's exact tests to determine the statistical significance of differences in prevalence. Comparisons of interest were all six pairwise comparisons between source/location for produce samples (ICLF, COF, farmers' markets, and supermarkets) and one each for reservoir (manure/compost vs animal pen) and soil (ICLF vs COF) samples.

Additional analyses were applied to test for differences in the distribution of STEC serovars and VFs (among positive samples) and associations between the occurrence of pathogens (*Listeria*, *Salmonella*, STEC, and generic *E. coli*) and the abundance of coliforms, as a potential indicator of pathogen occurrence (10). The previously cited study looked at the association between pathogen occurrence and generic *E. coli*; however, generic *E. coli* was often below the limit of detection, and that is why the relationship between pathogen prevalence and total coliforms was evaluated instead. For STEC serovars and VFs, Fisher's exact test was applied as an omnibus test of differences in distribution between samples collected from different sources/locations for each of five groups of sample types (produce, soil, animal pen, reservoir, and manure/compost). The number of sources/locations varied for each group of samples, with four for produce, two for soil, three for animal pen, two for reservoir (animal pen, manure/compost), and four for manure/compost. Associations between pathogen/generic *E. coli* occurrence and abundance of total coliforms were determined by fitting logistic regressions for each pathogen/generic *E. coli* with presence/absence of the pathogen in the sample as the response variable and log coliform count in the sample as the predictor variable. A positive (negative) estimate for the slope of the regression was interpreted as indicating a positive (negative) association between prevalence and abundance of coliforms, and the statistical significance of the association was determined by that of the estimated slope. Given potential differences by sample type, logistic regressions were applied separately for each of the three sample types (produce, soil, and reservoir) in addition to being pooled over sample type.

No adjustments were made for multiple comparisons, with *P*-values unadjusted for either the total number of comparisons overall or the total number of comparisons within different statistical analyses, for which there were varying numbers of comparisons. An alpha level of 0.05 was considered the minimum level for statistical significance, corresponding to a per-comparison type I error rate of 5% in the absence of adjustments for multiple comparisons. All calculations were performed using R 4.1.2.

## RESULTS

### Animal reservoir contamination (animal pen/BSAAOs) on ICLFs

#### Petrifilm microbial levels

High microbial concentrations were associated with ICLF animal pens and manures/composts (Fig. 1; Table 5). Of all tested samples, the highest APC and total coliform concentrations were from pig pens (10.76 and 8.70 log CFU/g, respectively). The prevalence of generic *E. coli* (84.68%) and total coliforms (99.19%) in animal pens was high. For BSAAOs, the highest APC (10.23 log CUF/g) and total coliform (7.90 log CFU/g) levels were found in cow/pig manure. Prevalence of generic *E. coli* and total coliforms in manure/compost was 49.58% and 99.16%, respectively. Generic *E. coli* and total coliforms went undetected in ICLF and COF well water; hence, pathogen prevalence in water was not evaluated.

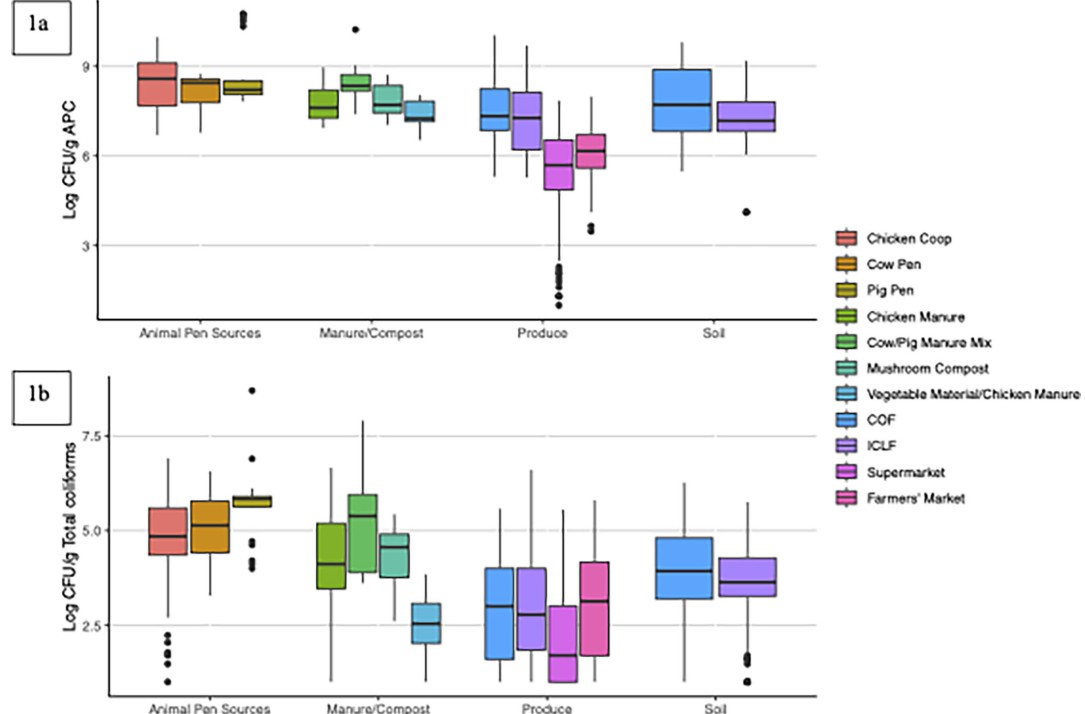

**FIG 1** Box plot of the log abundance for aerobic plate count (APC) (1a) and total coliforms (1b) of different sample types and locations. Log10 APC/Total Coliforms represents Log CFU/g values.

## Pathogen prevalence

ICLF animal pens were 3.23%, 0.81%, and 25.81% positive for *Salmonella*, *L. monocytogenes*, and STEC/VF-genes, respectively (Fig. 2). *Salmonella* was more prevalent in summer at 75% (3/4), and 50% (2/4) of isolations came from pig pens. *L. monocytogenes* was detected in one spring chicken-coop sample. STEC/VF genes were the most frequently isolated pathogen among animal pens with 62.5% (20/32) in summer. Isolations originated from chicken coops, cow pens, and pig pens at prevalences of 50% (16/32), 28.13% (9/32), and 21.88% (7/32), respectively. The most prevalent serovar and VF-gene were O103 (13.71%) and *stx*2/*eae* (4.84%), respectively (Table 6). *Salmonella*, *L. monocytogenes*, and STEC/VF-genes were observed in 1.68%, 0.84%, and 21.85% of ICLF-BSAAO (Fig. 2). *Salmonella* and *L. monocytogenes* were detected in fall chicken manure and spring cow/pig manure, respectively. Most STEC/VF genes were detected in winter and from BSAAOs containing chicken manure. The most prevalent serovar and VF-gene was O103 (9.24%) and *stx*1 (5.88%) (Table 6).

## Produce and soil contamination on ICLFs and COFs

### Petrifilm microbial levels

For ICLF produce, APC and total coliforms ranged from 5.27 (collard greens) to 9.70 (squash) log CFU/g and 1.00 (multiple) to 6.60 (chili pepper) log CFU/g, respectively (Fig. 1; Table 5). In contrast, COF produce APC and total coliforms ranged from 5.30 (collard greens) to 10.04 (bell pepper) log CFU/g and 1.00 (multiple) to 5.58 (collard greens) log CFU/g, respectively. COF produce had the maximum APC (10.04 log CFU/g), whereas ICLF produce had the greatest total coliform level (6.60 log CFU/g). Generally, COF produce average APC of 7.50 log CFU/g, which were comparable with those of ICLF produce (7.26 log CFU/g). However, among fall samples, COF produce (*n* = 150) average APC (7.50 log CFU/g) was significantly greater (*P* < 0.05) than the average ICLF produce (*n* = 66) APC of 6.97 log CFU/g. Moreover, ICLF produce generic *E. coli* and total coliform

**TABLE 5** Mean and range of log abundance for aerobic place count (APC) and total coliforms of different sample types and locations[a]

| | Mean APC (log CFU/g) | Range APC (log CFU/g) | Mean total coliforms (log CFU/g) | Range total coliforms (log CFU/g) | Total coliform prevalence (%) | Generic *E. coli* prevalence (%) |
|---|---|---|---|---|---|---|
| Produce | | | | | | |
| Integrated crop livestock farms (*n* = 257) | 7.26 | 5.27–9.70 | 2.95 | 1.00–6.60 | 87.94% | 6.61%[a] |
| Crop-only farms (*n* = 150) | 7.50 | 5.30–10.04 | 2.88 | 1.00–5.58 | 85.33% | 0.67%[b] |
| Farmer's markets (*n* = 230) | 6.12[a] | 3.48–7.99 | 2.99[a] | 1.00–5.80 | 85.65% | 3.48%[a] |
| Supermarkets (*n* = 273) | 5.50[b] | 1.00–7.85 | 2.03[b] | 1.00–5.56 | 65.93% | 0.00%[b] |
| Soil | | | | | | |
| Integrated crop livestock farms (*n* = 441) | 7.29[b] | 4.11–9.19 | 3.69[b] | 1.00–5.76 | 99.09% | 15.65%[a] |
| Crop-only farms (*n* = 150) | 7.83[a] | 5.48–9.81 | 3.90[a] | 1.00–6.27 | 99.33% | 5.33%[b] |
| Animal pen (*n* = 124) | 8.42 | 6.70– 10.76 | 5.00 | 1.00–8.70 | 99.19% | 84.68% |
| Chicken (*n* = 78) | 8.42 | 6.70–9.98 | 4.79 | 1.00–6.91 | 98.72% | 87.18% |
| Cow pen (*n* = 23) | 8.15 | 6.78–8.75 | 5.05 | 3.28–6.57 | 100% | 69.57% |
| Pig pen (*n* = 23) | 8.69 | 7.82–10.76 | 5.68 | 4.00–8.70 | 100% | 91.30% |
| Manure/compost (*n* = 119) | 7.82 | 6.53–10.23 | 4.14 | 1.00–7.90 | 99.16% | 49.58% |
| Chicken (*n* = 38) | 7.69 | 6.93–8.96 | 4.20 | 1.00–6.66 | 100% | 84.21% |
| Cow/Pig (*n* = 23) | 8.45 | 7.40–10.23 | 5.22 | 3.61–7.90 | 100% | 82.61% |
| Mushroom (*n* = 36) | 7.84 | 7.04–8.72 | 4.34 | 2.60–5.43 | 100% | 22.22% |
| Chicken/Veggie (*n* = 22) | 7.36 | 6.53–8.04 | 2.62 | 1.00–3.85 | 95.45% | 0% |

[a]Values with different letters are significantly different (p<0.05). Pairwise comparisons were performed on like samples between certain locations (integrated crop-livestock vs. crop-only farms and farmers' market vs. supermarket). Pairwise comparisons were not performed between types of animal pen and manure/compost samples.

detections, 6.61% and 87.94%, respectively, exceeded 0.67% and 85.33% of COF produce. The generic *E. coli* prevalence difference was significant (*P* < 0.05).

For ICLF soils, APC and total coliforms ranged 4.11–9.19 log CFU/g and 1.00–5.76 log CFU/g, respectively, whereas COF soils had APC and total coliforms from 5.48 to 9.81 log CFU/g and 1.00–6.27 log CFU/g. On average, COF soils (7.83 and 3.90 log CFU/g) had significantly higher (*P* < 0.05) APC and total coliforms than ICLF soils (7.29 and 3.69 log CFU/g), respectively. Similar to COF produce, COF soils had the maximum APC and total coliforms. Although COF soils had the maximum generic *E. coli* level, it was significantly more prevalent (*P* < 0.05) in ICLF soils (15.65%) compared with COF soils (5.33%). However, among fall samples, generic *E. coli* prevalence was 9.47% and 5.33% in ICLF (*n* = 95) and COF soils (*n* = 150), respectively, and the difference was not significant. Total coliform prevalence was 99.09% and 99.33% for ICLF and COF soils, respectively.

### Prevalence of pathogens

Overall, the prevalence of pathogens was higher in ICLF than in COF samples. *Salmonella*, *L. monocytogenes*, and STEC/VF-genes were detected in 1.67%, 1.98%, and 19.29% of ICLF samples, respectively, compared with 0.63%, 0.00%, and 11.88% of COF samples (Fig. 3). Like the overall prevalence, ICLF produce and soils were more contaminated with pathogens than their COF counterparts. *Salmonella* (0.39%) and *L. monocytogenes* (1.95%) were detected in ICLF produce, but the pathogens were not isolated from COF produce (Fig. 4). *Salmonella* was detected in late-season mustard greens. Most *L. monocytogenes* detections (four mustard greens and one squash) occurred in the fall (80%, 4/5). STEC/VF genes were detected significantly more (*P* < 0.05) in ICLF produce (13.62%) compared with COF produce (5.33%). Most ICLF detections occurred in fall at 68.57% (24/35) prevalence, and all COF isolations occurred during fall. For ICLFs, bell peppers and mustard greens had the most frequent isolations at a rate of 25.71% (9/35), whereas most COF isolations came from collard greens, 37.5% (3/8). The highest detected serovar for ICLF (5.84%) and COF produce (2.67%) was O103 (Table 6). The most prevalent VF gene for ICLF produce was *ehx*A (5.45%), and *stx*1 and *stx*2 (1.33% each) were the dominant VFs for COF produce.

*Salmonella*, *L. monocytogenes*, and STEC/VF-genes were found in 2.04%, 2.72%, and 20.86% of ICLF soils, respectively, compared with 1.33%, 0.00%, and 20.00% in COF soils

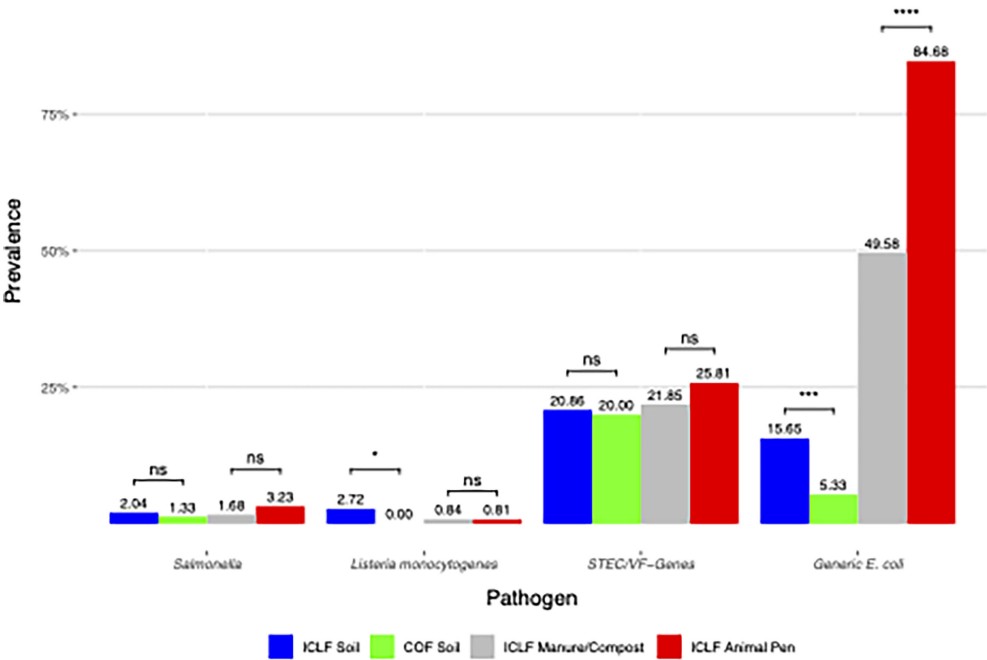

**FIG 2** Prevalence (%) and distribution of *Salmonella*, *L. monocytogenes*, Shiga toxin-producing *E. coli,* and virulence-factor genes (STEC/VF-genes) and generic *E. coli* in soil and animal reservoir samples obtained from integrated crop-livestock farms (ICLF) and crop-only farms (COF) on the Eastern Shore of Maryland. The asterisks (*) indicate levels of statistical significance where *, **, ***, and **** represent that *P*-values are *P* < 0.05, *P* < 0.01, *P* < 0.001, and *P* < 0.0001, respectively. Comparisons that were not significant are denoted by ns.

(Fig. 2). In ICLF soils, overall *L. monocytogenes* prevalence was significantly greater (*P* < 0.05) than that of COF soils, but there was no difference during the fall. In the fall subset of data, ICLF soils (*n* = 95) had a *Salmonella* prevalence (7.37%) significantly (*P* < 0.05) greater than that of COF soils as well. For *Salmonella*, most ICLF soil detections occurred in the fall and from soils that received a cow/pig manure amendment. The two COF soil detections occurred in fall. ICLF soil *L. monocytogenes* detections mainly occurred in spring and winter, each with prevalences of 41.67% (5/12), and 50.00% (6/12) came from mushroom compost amended soils. For STEC/VF genes, 53.26% (49/92) of detections were in summer, and 42.39% (39/92) originated from soils amended with mushroom compost. All COF soil detections occurred in fall. From ICLF soils, O103 (7.94%) and *stx*2 (3.63%) were the most prevalent STEC serovar and VF-gene (Table 6), respectively. For COF soils, the most dominant serovar and VF-gene were O103 (10.67%) and *ehx*A (6.00%), respectively.

## Produce contamination in farmers' market and supermarket retail environments

### *Petrifilm microbial levels*

For farmers' market produce, APC and total coliforms ranged from 3.48 (tomato) to 7.99 (squash) log CFU/g and 1.00 (multiple) to 5.80 (squash) log CFU/g, respectively. The supermarket produce had APC and total coliforms ranging from 1.00 (tomato) to 7.85 (kale) log CFU/g and 1.00 (multiple) to 5.56 log CFU/g, respectively (Fig. 1 and Table 5). On average, farmers' market produce had APC and total coliform levels of 6.12 and 2.99 log CFU/g, respectively, which were significantly greater (*P* < 0.05) than the supermarket produce levels of 5.50 and 2.03 log CFU/g. Furthermore, farmers' market produce had a

**TABLE 6** Prevalence and distribution of Shiga toxin-producing serovars and virulence-factor genes across various sample categories and types

| | O45 | O103 | O121 | O145 | O26 | O157 | O111 | Distribution (%) | stx1 | stx2 | eae | ehxA | Distribution (%) |
|---|---|---|---|---|---|---|---|---|---|---|---|---|---|
| **Produce** | | | | | | | | | | | | | |
| Integrated crop livestock farms (n = 257) | 1 | 15 | 0 | 4 | 3 | 6 | 1 | 12.99 (30/231) | 4 | 5 | 0 | 14 | 18.70 (23/123) |
| Crop-only farms (n = 150) | 2 | 4 | 0 | 0 | 0 | 1 | 0 | 3.03 (7/231) | 2 | 2 | 0 | 0 | 3.25 (4/123) |
| Farmer's markets (n = 230) | 0 | 0 | 0 | 0 | 0 | 0 | 0 | 0.00 (0/231) | 0 | 0 | 0 | 0 | 0.00 (0/123) |
| Supermarkets (n = 273) | 0 | 0 | 0 | 0 | 0 | 0 | 0 | 0.00 (0/231) | 0 | 0 | 1 | 0 | 0.81 (1/123) |
| **Soil** | | | | | | | | | | | | | |
| Integrated crop-livestock farms (n = 441) | 14 | 35 | 9 | 14 | 8 | 22 | 6 | 46.75 (108/231) | 10 | 16 | 12 | 10 | 39.02 (48/123) |
| Crop-only farms (n = 150) | 7 | 16 | 0 | 0 | 0 | 0 | 1 | 10.39 (24/231) | 2 | 6 | 0 | 9 | 23.82 (17/123) |
| Animal pen (n = 124) | 4 | 17 | 5 | 5 | 4 | 2 | 0 | 16.02 (37/231) | 3 | 6 | 6 | 4 | 15.45 (19/123) |
| Chicken (n = 78) | 3 | 7 | 4 | 0 | 2 | 1 | 0 | 7.36 (17/231) | 3 | 5 | 0 | 3 | 8.94 (11/123) |
| Cow pen (n = 23) | 0 | 7 | 1 | 4 | 0 | 0 | 0 | 5.19 (12/231) | 0 | 0 | 4 | 1 | 4.07 (5/123) |
| Pig pen (n = 23) | 1 | 3 | 0 | 1 | 2 | 1 | 0 | 3.46 (8/231) | 0 | 1 | 2 | 0 | 2.44 (3/123) |
| Manure/compost (n = 119) | 5 | 11 | 0 | 0 | 4 | 4 | 1 | 10.82 (25/231) | 7 | 3 | 1 | 0 | 8.94 (11/123) |
| Chicken (n = 38) | 1 | 5 | 0 | 0 | 2 | 1 | 1 | 4.33 (10/231) | 2 | 0 | 0 | 0 | 1.63 (2/123) |
| Cow/pig (n = 23) | 4 | 1 | 0 | 0 | 0 | 0 | 0 | 2.16 (5/231) | 0 | 1 | 0 | 0 | 0.81 (1/123) |
| Mushroom (n = 36) | 0 | 2 | 0 | 0 | 0 | 1 | 0 | 1.30 (3/231) | 0 | 0 | 1 | 0 | 0.81 (1/123) |
| Chicken/veggie (n = 22) | 0 | 3 | 0 | 0 | 2 | 2 | 0 | 3.03 (7/231) | 5 | 2 | 0 | 0 | 5.69 (7/123) |

higher generic *E. coli* and total coliform detection rate at 3.48% and 85.65%, respectively, compared with the 0.00% and 65.93% of supermarket produce. Farmers' market produce had a significantly greater ($P < 0.05$) generic *E. coli* prevalence.

### Prevalence of pathogens

*Salmonella* (kale) and *L. monocytogenes* (squash) were each prevalent in 0.43% of summer farmers' market produce, while undetected in supermarket produce. Only 0.37% of supermarket produce was positive for STEC/VF genes, but the pathogen was absent on farmers' market produce (Fig. 4). The STEC/VF-gene (*eae*) was detected in fall from kale (Table 6).

### Salmonella serotyping

Of the 19 *Salmonella* isolates, 13 were identified. The serovars were Bareilly (4), Newport (3), Javiana and Thompson (2 each), and Give and Rough O:Y:1, 5 (1 each). Bareilly was isolated from four ICLF soils, Newport from each one ICLF chicken manure, ICLF mustard green, and COF soil; Thompson from two ICLF pig pen samples; and Javiana from one ICLF cow pen and farmers' market kale. Regarding the six unidentified serovars, four were from ICLF soils, one from ICLF chicken manure, and one from COF soil.

### Relationship between pathogen prevalence and total coliforms

Among all data, there was a statistically significant ($P < 0.05$) relationship between three foodborne pathogens/generic *E. coli* and total coliform levels. There was also a statistically significant ($P < 0.05$) relationship between *L. monocytogenes* and total coliforms among produce. For produce, soils, and animal reservoirs (animal pen/BSAAO), there was a statistically significant ($P < 0.05$) relationship between generic *E. coli* prevalence and total coliforms as well.

### ICLF soil health

After the BSAAO application, all sites showed increases in nutrient concentrations and cation exchange capacity, with P, K, Ca, Mg, and cation exchange capacity increasing by 9%, 103%, 28%, 25%, and 29% on average, respectively (Table 7). Cation exchange capacity is the measure of a soil's ability to retain positively charged ions ($K^+$, $Ca^+$, $Mg^+$),

**FIG 3** The overall prevalence (%) of *Salmonella*, *L. monocytogenes*, Shiga toxin-producing *E. coli,* and virulence-factor genes (STEC/VF-genes), and generic *E. coli* in integrated crop livestock farm (ICLF), crop-only farm (COF), farmers market (FM), and supermarket (SM) environments on the Eastern Shore of Maryland. The asterisks (*) indicate levels of statistical significance where *, **, ***, and **** represent that *P*-values are *P* < 0.05, *P* < 0.01, *P* < 0.001, and *P* < 0.0001, respectively. Comparisons that were not significant are denoted by ns.

providing insight into nutrient-retention capacity (44, 45). Soil organic matter levels increased at ICLF-A (4%) and ICLF-C (13%) while remaining consistent on ICLF-B. Soil organic matter is the decomposed organic materials (animal, plant, soil) in the soil, which benefit soil structure, nutrient cycling, and water retention, and reduce erosion (46, 47). Only minor changes in soil pH were observed.

## DISCUSSION

### Animal reservoirs increase ICLF contamination risks

#### *Animal pen*

The ICLFs sampled in this study were practicing farms that are exempt from the Food Safety Modernization Act Produce Safety Rule and other standardized regulations; hence, contamination results reflect real-world practices of local, small-scale ICLFs. Animal pens were the most frequently contaminated samples from this study, with 29.84% determined positive for at least one pathogen. *Salmonella* and STEC/V -genes were detected in animal pens for chickens, cows, and pigs, whereas *L. monocytogenes* was only detected in chicken coops. Multiple species of animals were present on each sampled ICLF, and two of them owned swine. At times, different species occupied the same space or were in proximity to each other, which could facilitate the spread of foodborne pathogens. Ownership of swine and shared barns has previously been indicated as a significant risk factor for a pathogen-positive sample (12, 48). Small operations may not have the space or the facilities to separately lodge all animals. Therefore, the transmission between pathogens and animal hosts needs to be addressed (12). Similar studies on practicing and model ICLFs reported *Salmonella* and STEC prevalences up to 18.6% in

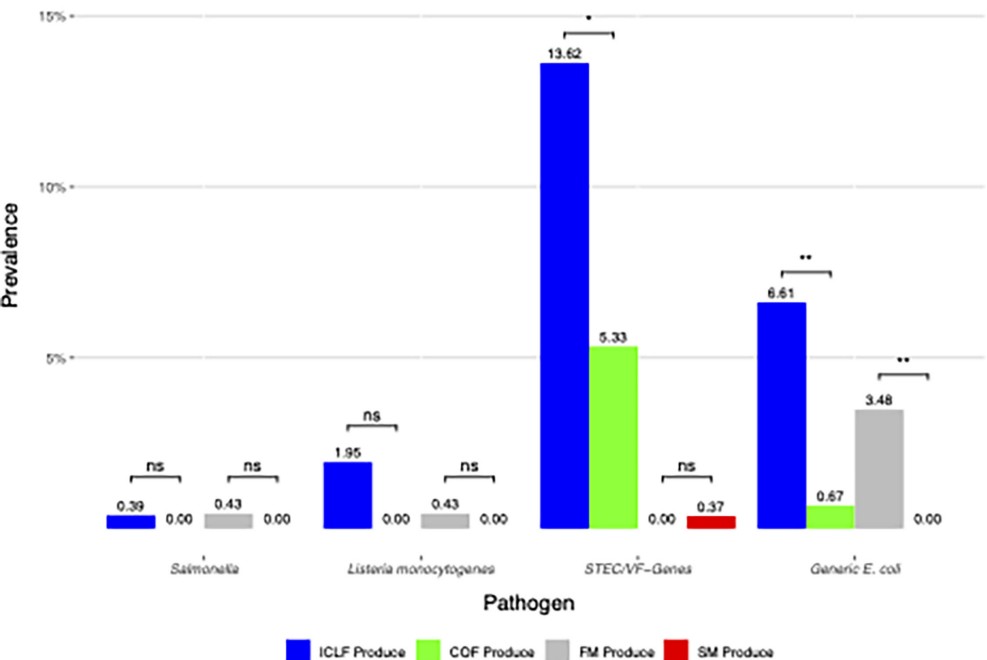

**FIG 4** Prevalence (%) of *Salmonella*, *L. monocytogenes*, Shiga toxin-producing *E. coli,* and virulence-factor genes (STEC/VF-genes) and generic *E. coli* in produce samples obtained from integrated crop livestock farms (ICLF), crop-only farms (COF), farmers markets (FM), and supermarkets (SM) on the Maryland Eastern Shore. The asterisks (*) indicate levels of statistical significance where *, **, ***, and **** represent that *P*-values are *P* < 0.05, *P* < 0.01, *P* < 0.001, and *P* < 0.0001, respectively. Comparisons that were not significant are denoted by ns.

animal-rearing (cattle, poultry, goat, sheep, swine) facilities, e.g., grass, feces, feed, hide, water, and bedding (16, 48–50). *L. monocytogenes* has also been isolated from animal rearing facilities (forage, fecal, soil, slurry, etc.) associated with cattle, poultry, and swine (51–53). Although livestock are absent, COF agricultural water, soil, and crops can be contaminated by animal (domestic/wildlife) feces, like ICLFs. Belias et al. (54) found that wildlife feces were positive (29%) for *L. monocytogenes* on a New York produce farm. Multiple studies have described pathogen contamination risks associated with animal reservoirs; hence, it is imperative that ICLF farmers follow Good Agricultural and Good Handling Practices with animal-rearing facilities near crop soils and take steps to reduce wild animal intrusion.

### BSAAO – manures/composts

The increased microbial contamination of ICLF animal reservoirs is likely attributable to foodborne pathogens being naturally associated with the gastrointestinal tracts of farm animals (55). In our study, 24.37% of BSAAOs were positive for at least one tested pathogen, and contamination was present in BSAAOs representing various animal species and management (aged on farm, commercial compost) practices. Although this study had low *Salmonella* and *L. monocytogenes* prevalence, STEC/VF-gene prevalence was high. Compared with our study, Ramos et al. (10) found higher rates of *Salmonella* (7.3%) and *L. monocytogenes* (3.9%) in BSAAOs, but STEC/VF-genes (10.3%) at a lower rate. In Georgia and Ohio, *Salmonella* (2%), *L. monocytogenes* (42%), STEC (15%), *Campylobacter* (24%), and *Arcobacter* (33%) were isolated from all amendment samples (56). These isolations originated from dairy and poultry BSAAOs, but no pathogens were detected in green (non-animal) composts (56). In Maryland and Washington D.C., ICLFs, composts (3.52%) were positive for *Salmonella*, but diarrheagenic *E. coli* went undetected

**TABLE 7** Integrated crop-livestock farm (ICLF) basic soil fertility test results before and after biological soil amendment of animal origin incorporation

| Source | ICLF A (n = 3) | | ICLF B (n = 3) | | ICLF C (n = 2) | | All samples (n = 8) | |
|---|---|---|---|---|---|---|---|---|
| | Before | After | Before | After | Before | After | Before | After |
| pH | 5.9 | 5.8 | 5.4 | 5.6 | 7.2 | 7.3 | 6.0 | 6.1 |
| Buffer pH | 6.6 | 6.5 | 6.6 | 6.7 | 7.0 | 7.0 | 6.7 | 6.7 |
| Phosphorous, ppm | 358 | 370 | 558 | 594 | 74 | 141 | 363 | 397 |
| P saturation ratio | 65 | 70 | 113 | 125 | 22 | 44 | 73 | 84 |
| Potassium, ppm | 125 | 377 | 105 | 130 | 106 | 137 | 111 | 224 |
| Calcium, ppm | 1,200 | 1,353 | 489 | 614 | 1,641 | 2,276 | 1,021 | 1,307 |
| Magnesium, ppm | 99 | 135 | 52 | 53 | 222 | 289 | 114 | 143 |
| Cation exchange capacity (meq/100 g) | 8.9 | 11.0 | 4.6 | 5.2 | 10.8 | 14.7 | 7.6 | 9.8 |
| Organic matter, % | 5.1 | 5.3 | 2.1 | 2.1 | 3.9 | 4.4 | 3.5 | 3.9 |

(49, 50). Composts produced using validated processes inactivate pathogens, but on ICLF-A, the commercially produced mushroom compost was stored unprotected (outdoors, uncovered), exposing the amendment to recontamination. ICLF-B and ICLF-C utilized minimally managed, aged manures, not a validated composting process (15).

The relationship between pathogens and the BSAAO source can be a crucial element in identifying ICLF contamination risks. *Salmonella* detections in the current study originated from chicken manure, a common *Salmonella* reservoir (10, 16, 27, 57–59). The sole *L. monocytogenes* isolation occurred from cow/pig manure mix, which sources previously reported (10, 15, 60). As in comparable studies, we isolated STEC/VF genes from BSAAOs containing chicken, cow, horse, and pig manure (10, 56). Development and dissemination of achievable best practices for storing, managing, and applying BSAAOs is essential to reducing pathogen loads on ICLFs.

## Water

In this study, the absence of generic *E. coli* and total coliforms in well water samples does not indicate the absence of harmful organisms; instead, they may be below the analytical detection limit (one organism per 100 mL). Similar to our study, Cheong et al. (61) did not detect generic *E. coli* in irrigation water. Ramos et al. (10) did not detect pathogens in ICLF water; however, they did isolate generic *E. coli* from 20.5% of water samples, with most occurring in surface water. Several studies have isolated *Salmonella*, *L. monocytogenes*, and STEC from water sources (non-irrigation/irrigation), indicating contamination risks associated with water reservoirs (27, 28, 35, 57, 62–65). A New York produce farm's agricultural water samples were positive (48%) for *L. monocytogenes*, possibly from dairy farm runoff (54). Although COFs lack farm animals, nearby animal operations and wildlife can contaminate agricultural water, soil, and produce. The lack of generic *E. coli* and total coliform detections could also be due to the tested water being sourced from wells. Multiple studies have found lower contamination associated with well or municipal water (57, 62–64).

## ICLF vs. COF produce contamination risks

In the current study, pathogens were isolated from ICLF produce harvested from soils that received different manure-based amendments. *Salmonella*-positive produce came from chicken manure-amended soils, and *L. monocytogenes* positives were from soils amended with chicken, cow, or pig manure. For STEC/VF-genes, positive samples were harvested from soils that were amended with BSAAOs containing these same manures and horse manure. Multiple studies have detected foodborne pathogens in produce harvested from manure (poultry, ruminant, pig, and horse) amended soils (10, 14, 17, 57, 66). Furthermore, multiple produce types from farms were positive for pathogens in this study. Most detections occurred in mustard greens, bell peppers, and collard greens. Although mustard and collard greens are typically cooked before consumption, frequent contamination reflects the proximity of these and similar harvested commodities to soil.

Similar to previous studies on practicing farms that evaluated contamination differences between ICLFs and COFs, the present study found that overall pathogen and generic *E. coli* prevalence were greater among ICLF produce (15.95% and 6.61%) than COF produce (5.33% and 0.67%), respectively. Peng et al. (2) found 6.5% of ICLF produce (e.g., leafy greens, tomatoes, green peppers, etc.) to be more contaminated with *Salmonella* than organic (0.7%) and conventional (0.6%) COF produce, respectively. Ramos et al. (10) did not detect *Salmonella* or STEC/VF genes on ICLF produce but isolated *L. monocytogenes* from two produce samples cultivated in different fields (unamended, cattle manure-amended). In experimental fields, *Salmonella*, *L. monocytogenes,* generic *E. coli*, and STEC O157:H7 were not detected on produce from manure-amended, conventionally fertilized, grazed, and control soils, which indicates that pathogens have gone undetected in ICLF environments (55, 57, 61). However, experimental fields are in a controlled setting where variables are kept consistent, which does not reflect the environment of practicing farms with different management strategies. For example, in the current study, each farm managed its BSAAOs differently, which can affect pathogen prevalence, and one farm left the amendment exposed to recontamination, increasing risks of pathogen transfer between manure amendments and soil and produce.

Similar to previously published studies, *Salmonella* and *L. monocytogenes* isolations were relatively low among ICLF produce (0.39% and 1.95%, respectively) and were undetected in COF counterparts. In poultry and cow manure-amended soils, Ekman et al. (14) did not detect *Salmonella* or *Listeria* spp. in mature lettuce but found *Listeria* in immature plants. Following standardized withholding periods and harvesting healthy plants are important in generating safe products. *Salmonella* went undetected in Mid-Atlantic region organic and conventional farms (29). In the current study, STEC/VF-genes were detected significantly ($P < 0.05$) more frequently among ICLF (13.62%) than COF produce (5.33%). In Maryland and Washington D.C., Aditya et al. (50) found organic ICLF produce (51.28%) positive for diarrheagenic *E. coli* virulent types. Lower STEC prevalences have been reported, where one study found 2.68% of sustainably farmed produce positive for STEC, and another study on small-to-medium, organic, and conventional farms did not detect the pathogen on produce (17, 29). However, the latter study did detect *stx*1 and *stx*2 in one tomato. In our study, generic *E. coli*, a common STEC indicator, was detected significantly greater ($P < 0.05$) among ICLF produce than COF produce. Previously published data on fecal contamination in organic and conventional farms are inconsistent with our findings. Ekman et al. (14) detected generic *E. coli* on lettuce from unamended and amended (cow manure/poultry litter) soils, but the only detection on mature lettuce was from unamended plots. Pagadala et al. (29) found that conventional farm produce had greater generic *E. coli* concentrations and prevalence (significant, $P < 0.05$) than organic counterparts, indicating contamination vectors other than BSAAOs. Although ICLF produce had increased pathogen and generic *E. coli* prevalence, COF produce, along with COF soils, had higher average APC than their counterparts. Chemical restrictions and BSAAOs increase biodiversity, which may lead to reduced ICLF produce and soil APC due to competition, parasitism, and predation of microorganisms (67–69). Analyses are needed to evaluate how produce and soil microbiomes influence contamination risks in preharvest environments. Multiple reports of pathogens and indicator organisms in both ICLF and COF produce indicate the food safety risks regarding the growing and harvesting of crops associated with organic and conventional farming practices that need to be further addressed.

## ICLF vs. COF soil contamination risks

Similar to produce, ICLF soils (2.04% and 2.72%) were more contaminated with *Salmonella* and *L. monocytogenes* than COF soils (1.33% and 0.00%), respectively. *Salmonella* was significantly more prevalent in ICLF soils (7.37%) than in COF soils (1.33%) in the fall subsets of data. *Salmonella* has been associated with increased temperatures, but suitable climatic conditions and cross-contamination through field activity may

have caused this significant difference (28, 57, 63). Field activity could include farmers moving between animal pens and crop fields or U-Pick activities, which ICLF practiced, which can increase pathogen transmission. Among ICLF detections, *Salmonella* came from soils that received BSAAOs containing cow, pig, and horse manure, whereas *L. monocytogenes* was isolated from soils that received amendments containing the same manures plus chicken manure. *Salmonella* and *L. monocytogenes* prevalence in our study was low and is likely due to infrequent detections in BSAAOs. Across multiple regions, *Salmonella* and *L. monocytogenes* were detected in manure-amended soils in prevalences up to 14.32% and 19%, respectively (10, 49, 56). In a study analyzing the integration of pastured meat chickens into organic vegetable production, the only *Salmonella* detection occurred in the vegetable-cover crop rotation instead of rotations involving poultry, which demonstrates pathogen contamination is not unique to BSAAO-amended soils (55). Studies conducted on Virginia and New York produce farms found *Salmonella and L. monocytogenes* in prevalences up to 5% and 13%, respectively (27, 62–65).

STEC/VF genes were detected more frequently than the other pathogens and generally had a similar prevalence in ICLF (20.86%) and COF soils (20.00%). Farming practices or environmental factors may provide a more suitable environment for the survival and transmission of STEC/VF-genes since Ramos et al. (10) also found a higher prevalence of the pathogen compared with *Salmonella* and *L. monocytogenes*. On ICLFs, in this study, STEC/VF-genes were prevalent in three different BSAAOs, which may be the reason for their increased detection in soils and produce, indicating the contamination risks associated with the survival and transmission of pathogens through BSAAOs. Infrequent STEC detections in BSAAO (poultry manure) amended soils were reported on Georgia and Ohio farms (56). Mid-Atlantic region ICLF soils were 5.12% positive for major virulent *E. coli* (50). STEC/VF-gene contamination has been described in soils where BSAAOs were not applied. Non-O157 STEC was detected in one (0.1%) fallow soil sample but went undetected in grazed samples. However, this same study isolated generic *E. coli* in 56.9%, 23.3%, and 30.6% of grazed, non-grazed, and fallow soils (61). Another rotational study detected STEC O157:H7 in 50%, 33%, and 42% of vegetable-cover crop (no amendment), vegetable-cover crop-poultry, and vegetable-poultry-cover crop-treated soils, respectively (55). Organic farms rely on biological soil amendments, and Pagadala et al. (29) found generic *E. coli* to be more prevalent in conventional (12.5%) farm soils, despite the lack of BSAAOs and animal rearing facilities, than in organic (4.8%) farm soils. Contrastingly, our study found generic *E. coli* to be significantly more prevalent in ICLF soils (15.65%) than in COF soils (5.33%). However, the difference was smaller in fall subsets of data, where ICLF soils had a generic *E. coli* prevalence of 9.47% compared with the 5.33% of COF soils. In our study, 55.56%, 8.33%, 20.56%, and 17.39% of ICLF control soils, which did not receive BSAAOs, were positive for *Salmonella*, *L. monocytogenes*, STEC/VF-genes, and generic *E. coli*, respectively. A tracing analysis was not performed to identify contamination sources, but the likely vectors were farm, domestic, or wild animal reservoirs. Transmission could have occurred through farmer activity or weather events. Multiple reports of pathogen presence in agricultural soils, whether organic or conventional, indicate that BSAAO and farm animals are not the sole soil contamination vectors.

## Farmers' market vs. supermarket contamination risks

Total pathogen and generic *E. coli* prevalence and average APC and total coliforms were greater in farmers' markets, indicating that farmers' market produce was more contaminated than supermarket produce in this study. Similarly, Peng et al. (2) found *Salmonella* at a higher rate in farmers' market produce (16.8%) when compared with produce obtained from organic (6.1%) and conventional (0%) retail stores in Maryland and Washington D.C. Higher contamination levels were also associated with Florida farmers' market, where these samples had pathogen (*L. monocytogenes*, *Salmonella*), generic *E. coli*, and total coliform prevalences greater than supermarket produce (9). No human pathogens (*Shigella*, *Salmonella*, STEC) were isolated from cilantro, onions, and

peppers obtained from national chains or locally owned stores and farmers' markets across multiple states. However, average APC populations of locally owned stores and farmers' market produce were significantly greater than those of national chain store samples. Farmers' market produce also had slightly higher average total coliforms (70). In Maryland and Washington D.C., virulent *E. coli* and *Campylobacter* were not detected in produce collected from farmers' markets and organic and conventional retail stores (11, 50). Generally, from the present and cited studies, higher contamination levels are associated with farmers' markets and other unique (organic and locally owned) retail venues when compared with traditional (conventional, supermarket) retail venues. Traditional retail venues typically receive products from large-scale, conventional farms, and both settings are more stringently regulated and likely have the knowledge and capability to implement proper Good Agricultural and Good Handling Practices to reduce contamination risks. Produce tends to be washed with water-containing sanitizer before delivery to supermarkets. Whereas farmers' markets and their likely small-scale, local suppliers are less stringently regulated and may not have the necessary resources and training to incorporate proper food safety practices, which increases produce contamination and consumer foodborne illness risks.

In our study, pathogen and fecal contamination in farmers' market and supermarket produce were relatively low, which may indicate proper farm and produce handling in these Maryland pre- and post-harvest environments. Each pathogen was detected once. *Salmonella* and *L. monocytogenes* were found in one farmers' market produce each, whereas a STEC-VF gene (*eae*) was found in one supermarket produce. Generic *E. coli* was isolated from farmers' market produce (3.48%) and went undetected in supermarket produce. Pathogen prevalence in this study was similar and noticeably lower when compared with studies conducted in like environments. In West Virginia and Kentucky farmers' markets, Li et al. (71) detected *Salmonella* at a higher rate compared with our study (16.03%), but *L. monocytogenes* had a low prevalence of 1.89%. Produce from Central Virginia farmers' markets was positive for *Campylobacter*, generic *E. coli*, *L. monocytogenes*, and *Salmonella* at a prevalence of 8.7%, 9.4%, 2.2%, and 0.0%, respectively (5). Pires et al. (72) did not isolate *Salmonella* from farmers' market produce but detected generic *E. coli* in 31.3% of produce in Northern California. This study also detected *Salmonella* in 1.8% of animal products (raw beef, chicken, and pork). Although generic *E. coli* prevalence was greater than in our study, low *Salmonella* prevalence in California may be due to sampling at certified farmers' markets, which are likely subject to stricter food safety regulations (72). Increased pathogen prevalence associated with animal products at farmers' markets is concerning, as these products can act as major contamination vectors for produce. In the current study, meat and produce products were available, and domesticated pets were present as well, both increasing contamination risks. Furthermore, some vendors stored produce in ice coolers, whereas others displayed foods on tables that were not refrigerated. Most foodborne pathogens proliferate at 10°C or greater; improper temperature storage and display of foods risk contact contamination between animal products, produce, vendors, and consumers (5).

Korir et al. (73) observed a low prevalence (<0.5%) of *Salmonella*, *L. monocytogenes*, and STEC O157:H7 in supermarket produce on the Maryland Eastern Shore. In small, independently owned and large-chain supermarket stores, *Salmonella* (0.0%) and generic *E. coli* (4.9%) were detected in produce (74). Across multiple states, previous studies described pathogen and indicator organism prevalence on produce obtained from various (organic, conventional, ethnic, etc.) retail stores (75, 76). Although more stringently regulated than farmers' markets, contamination risks are associated with supermarket produce. Among all data, the likelihood of pathogen presence increased as total coliforms increased. Ramos et al. (10) reported a similar trend between pathogens and generic *E. coli*. These findings indicate that pathogen prevalence increases when sanitary conditions decline.

## BSAAO improved ICLF soil health and fertility

The benefits of animal manure for crop fertilization and soil health are widely recognized as essential for organic farms (10, 44, 77). However, organic crop production often depends on intensive tillage for weed management and residue incorporation, practices that can degrade soil organic matter (78, 79). Soil organic matter, a critical soil health indicator, is strongly linked to biological, chemical, and physical soil health metrics (80). In this study, soil organic matter was maintained or increased across all sites, likely due to external carbon inputs from BSAAOs. This aligns with numerous studies attributing soil organic matter increases in organic systems to external carbon contributions (81–84). Thus, integrating BSAAOs can help offset the ramifications of tillage on soil health by sustaining or enhancing soil organic matter. Organic farms frequently utilize BSAAOs for crop fertilization, as they are a valuable source of essential nutrients (44, 85). However, improper manure management and over-application can lead to environmental concerns, particularly excess N and P runoff, which threaten water quality in the Chesapeake Bay watershed (86, 87). Although BSAAO application increased essential plant nutrient concentrations across all sites, Mehlich-3 extractable P levels were optimal (51–100 ppm) or excessive (101+ ppm) prior to application, indicating that additional manure application was not advisable (88). Notably, before the BSAAO application, ICLF-A (358 ppm) and ICLF-B (558 ppm) had extremely high soil test P levels (Table 7). Furthermore, their P saturation ratios—representing the proportion of phosphorus in the soil relative to its total retention capacity—exceeded 50, a threshold identified by the University of Delaware Soil Testing Laboratory as indicating a very high risk of P loss (89). This illustrates the importance of soil sampling for effective nutrient management to minimize agricultural impacts on water quality. However, despite the high soil test P concentrations, some crops, such as tomatoes and potatoes, may still exhibit yield responses to P fertilizer (88).

This study primarily focused on the food safety aspects of integrating BSAAOs in fresh produce systems. Limited samples and fertility analyses conducted within a week of BSAAO application were insufficient to capture the full effects of a slow-release fertilizer like manure on soil health (44, 90). Despite limitations, observed trends aligned with previously reported literature, highlighting the importance of soil sampling before BSAAO application for effective nutrient management.

In conclusion, study findings show greater contamination risks associated with ICLF and farmers' market produce compared with COF and supermarket counterparts, although pathogens were present in each environment. The increased contamination risks of ICLFs and farmers' markets are likely attributable to the described practices that are unique to each environment. Furthermore, soil health results indicated that BSAAOs can improve soil health parameters on small-scale ICLFs, but soil testing is critical for effective nutrient management. The results are unique to study design, farmer and vendor management practices, and geographical and meteorological conditions. For example, one ICLF experienced a flooding event in crop soils less than 9.144 m from a chicken coop. Although no products were marketed, samples post-flood revealed *Salmonella* (1) and *L. monocytogenes* (4) on mustard greens, and STEC/VF-genes in chicken coop (2) and soil (2) samples. Extreme weather events and soil proximity to animal-rearing facilities show the need to plan emergency management strategies. Statistical significance of results reported here was also based on controlling type I error at the level of each individual comparison or test. Adjustments to control type I error family-wise were not applied to limit the frequency of type II error (91). A total of 130 statistical comparisons were performed, not all of which were presented here due to word limitations and relevancy, and 66 were found to be statistically significant ($P < 0.05$) compared with 7 that would be expected by chance if the universal null hypothesis were true.

Foodborne illness is a recurring problem that negatively affects consumer health and producer profits. This study is limited in that results are unique to the studied locations and region and that farm management and produce handling practices vary.

Among operating ICLFs, COFs, farmers' markets, and supermarkets, there are uncontrolled variables that bias contamination comparisons. However, the results presented in this paper detail how these variables can influence pathogen prevalence, and the results are generalizable in that the practices associated with each environment can help explain differences in contamination. Thus, continued research in real-world settings on pathogen persistence and mitigation strategies is essential for ICLFs and farmers' markets in other regions. Proper dissemination of food safety standards regarding pre- and post-harvest practices (BSAAO management, animal and crop field proximity, withholding periods, and sanitation) to farmers, vendors, and relevant parties is vital to safe, fresh produce production.

## ACKNOWLEDGMENTS

First, we would like to thank the farmers and farmers' market vendors for their cooperation and participation throughout this study. We acknowledge Joan Meredith, Barbara Osei TuTu, Annette Kenney, Edwina Barnett, and Enrique Escobar for their help and support in field and experimental work. We would also like to extend gratitude to Mohammad Ali for facilitating the enrollment of farmers.

The author(s) declare financial support was received for the research, authorship, and/or publication of this article. This project was funded by USDA-NIFA, Organic Transitions Program grant no. 2018511062880, USDA NIFA CBG 2020-38821-31082, USDA NIFA CBG 5–208070, USDA Evans Allen, and USDA ARS.

B.G.: Writing – original draft, Investigation, Methodology, Data curation, Validation, Visualization. P.M.: Writing – review & editing, Conceptualization, Methodology, Funding acquisition, Validation, Visualization. A.P.-D.: Writing – review & editing, Investigation, Methodology. M.S.: Writing – review & editing, Investigation, Methodology, Supervision, Visualization. J.B.: Writing – original draft, Writing – review & editing, Data curation, Software, Formal Analysis. J.H.: Writing – original draft, Writing – review & editing, methodology. F.H.: Writing – review & editing, Funding acquisition, Visualization. C.K.: Writing – review & editing, Conceptualization, Funding acquisition, Project administration. D.B.: Writing – review & editing, Conceptualization, Funding acquisition, Project administration. S.P.: Writing – original draft, review & editing, Conceptualization, Funding acquisition, Methodology, Project administration, Resources, Supervision, Validation, Visualization.

## AUTHOR AFFILIATIONS

[1]Department of Agriculture, Food and Resource Sciences, University of Maryland Eastern Shore, Princess Anne, Maryland, USA
[2]USDA-ARS-Sustainable Agricultural Systems Laboratory, Beltsville, Maryland, USA
[3]U.S. Food and Drug Administration, College Park, Maryland, USA
[4]Eastern Shore Agricultural Research and Extension Center, Virginia Tech, Painter, Virginia, USA
[5]Agricultural Research Station, Virginia State University, Petersburg, Virginia, USA
[6]Department of Animal and Avian Sciences and Center for Food Safety and Security Systems, University of Maryland, College Park, Maryland, USA

## AUTHOR ORCIDs

Brian Goodwyn  http://orcid.org/0009-0008-9151-0458
Salina Parveen  http://orcid.org/0000-0002-1917-3847

## FUNDING

| Funder | Grant(s) | Author(s) |
|---|---|---|
| USDA NIFA | 20185110628802018511062880, 2020-38821-31082, 5-208070 | Debabrata Biswas |
| | | Salina Parveen |

| Funder | Grant(s) | Author(s) |
|--------|----------|-----------|
| | | Patricia Millner |
| | | Fawzy Hashem |
| | | Chyer Kim |

## AUTHOR CONTRIBUTIONS

Brian Goodwyn, Data curation, Investigation, Methodology, Validation, Visualization, Writing – original draft | Patricia Millner, Conceptualization, Funding acquisition, Methodology, Validation, Visualization, Writing – review and editing | Anuradha Jeewantha Punchihewage-Don, Investigation, Methodology, Writing – review and editing | Melinda Schwarz, Investigation, Methodology, Supervision, Visualization, Writing – review and editing | John Bowers, Data curation, Formal analysis, Software, Writing – original draft, Writing – review and editing | Joseph Haymaker, Methodology, Writing – original draft, Writing – review and editing | Fawzy Hashem, Funding acquisition, Visualization, Writing – review and editing | Chyer Kim, Conceptualization, Funding acquisition, Project administration, Writing – review and editing | Debabrata Biswas, Conceptualization, Funding acquisition, Project administration, Writing – review and editing | Salina Parveen, Conceptualization, Funding acquisition, Methodology, Project administration, Resources, Supervision, Validation, Visualization, Writing – original draft, Writing – review and editing

## DATA AVAILABILITY

The original contributions presented in the study are included in the article/Supplementary material, further inquiries can be directed to the corresponding author.

## ADDITIONAL FILES

The following material is available online.

Open Peer Review

**PEER REVIEW HISTORY (review-history.pdf).** An accounting of the reviewer comments and feedback.

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
