## [Reviewer comments · Microbiology Spectrum]

Microbiology Spectrum

Integrated Crop-Livestock Farming Systems Influence the Incidence of *Salmonella*, *Listeria monocytogenes*, Shiga toxin-producing *E. coli*, and Indicator Bacteria on Fresh Produce

Brian Goodwyn, Patricia Millner, Anuradha Punchihewage-Don, Melinda Schwarz, John Bowers, Joseph Haymaker, Fawzy Hashem, Chyer Kim, Debabrata Biswas, and Salina Parveen

Corresponding Author(s): Salina Parveen, University of Maryland Eastern Shore

Review Timeline:

Submission Date:	March 22, 2025
Editorial Decision:	May 29, 2025
Revision Received:	July 3, 2025
Editorial Decision:	July 26, 2025
Revision Received:	September 24, 2025
Accepted:	September 28, 2025

Editor: Dan Li

Reviewer(s): Disclosure of reviewer identity is with reference to reviewer comments included in decision letter(s). The following individuals involved in review of your submission have agreed to reveal their identity: Stanley Chen (Reviewer #2)

Transaction Report:

DOI: <https://doi.org/10.1128/spectrum.00862-25>

Re: Spectrum00862-25 (**Integrated Crop-Livestock Farming Systems Influence the Incidence of *Salmonella*, *Listeria monocytogenes*, Shiga toxin-producing *E. coli*, and Indicator Bacteria on Fresh Produce**)

Dear Dr. Salina Parveen:

Thank you for the privilege of reviewing your work. Below you will find my comments, instructions from the Spectrum editorial office, and the reviewer comments.

Revision Guidelines

Sincerely,
Dan Li
Editor
Microbiology Spectrum

Reviewer #1 (Comments for the Author):

I question whether comparing separate ICLFs and COFs-each with its own sewage lines, drainage systems, and potential background contamination-might confound the apparent effects of animal-based soil amendments; specifically, could unmonitored factors such as leaking septic or manure lagoons, farm-specific infrastructure, or wildlife intrusion have driven the differences in pathogen prevalence rather than the use of BSAAOs alone? Additionally, lines 34-35 and line 451 require

correction. While soil fertility was appropriately tested pre- and post-manure application, the absence of parallel pathogen measurements on the same plots leaves causality unestablished. It is also unclear why well water was screened only for generic *E. coli*/coliforms and not for *Salmonella* or *Listeria*, particularly given water's known role in pathogen dissemination. Moreover, the use of multiple Fisher's exact tests and ANOVAs without adjustment for the total number of comparisons raises the risk of inflated Type I errors, suggesting that some reported "significant" differences may be false positives. Finally, incorporating antimicrobial-resistance profiling for the detected pathogens would substantially enhance the study's relevance to public health.

Reviewer #2 (Comments for the Author):

The manuscript is well written. However, the biggest issue is the stat analysis in the result section and the way they are presented in tables/figures, i.e. no stat indications of significance between sample groups. P values are only mentioned in the body of text, but are missing in all tables/figures.

Response to the Reviewers' comments

Thank you very much for taking your valuable time to read and give feedback to develop our manuscript. All comments have been addressed and corresponding changes are marked and highlighted with track changes. Also, line numbers in the responses indicate line numbers with re-submitted clean version of the manuscript.

Reviewers' Comments

Reviewer #1 (Comments for the Author):

Comment 1:

I question whether comparing separate ICLFs and COFs—each with its own sewage lines, drainage systems, and potential background contamination—might confound the apparent effects of animal-based soil amendments; specifically, could unmonitored factors such as leaking septic or manure lagoons, farm-specific infrastructure, or wildlife intrusion have driven the differences in pathogen prevalence rather than the use of BSAAOs alone?

Response 1:

We maintain that comparing ICLFs and COFs is feasible because this research focuses on the contamination differences and risks associated with each of these two environments. While it is possible that other mentioned, unmonitored factors could contribute to differences in pathogen prevalences within the two environments, we acknowledged that BSAAOs are a likely contamination vector, and that other unique factors, such as wildlife, regions, and farm management, can influence pathogen prevalence as well (lines 500 – 504 and 596 – 599).

Comment 2:

Additionally, lines 34-35 and line 451 require correction. While soil fertility was appropriately tested pre- and post-manure application, the absence of parallel pathogen measurements on the same plots leaves causality unestablished.

Response 2:

In this study, the same plots were used for the soil fertility testing and the soil microbial (total aerobic bacteria, generic *E. coli*, total coliforms, *Salmonella*, *Listeria monocytogenes*, and shiga toxin-producing *E. coli*) analyses. Regarding pre- and post-manure application, the same composite soil samples were used for the soil fertility testing and the microbial analyses.

Comment 3:

It is also unclear why well water was screened only for generic *E. coli*/coliforms and not for *Salmonella* or *Listeria*, particularly given water's known role in pathogen dissemination.

Response 3:

We tested the irrigation water samples (all from well water) for generic *E. coli* and determined that all samples were negative. Therefore, we did not pursue analysis of the same samples and sources for pathogenic bacteria.

Comment 4:

Moreover, the use of multiple Fisher's exact tests and ANOVAs without adjustment for the total number of comparisons raises the risk of inflated Type I errors, suggesting that some reported "significant" differences may be false positives.

Response 4:

While it is true that some differences reported as significant may be type I errors it is also true that each individual comparison has a probability of type I error of 5%. We did not apply standard adjustments (e.g., Bonferroni) for multiple comparisons for several reasons. Foremost, such adjustments increase the probability of type II error, and the universal null hypothesis is not a reasonable expectation, based on previous studies reported in the literature. We do agree that absent any such adjustments the total number of comparisons merits discussion and have revised the manuscript accordingly (please see lines 656 - 662).

Comment 5:

Finally, incorporating antimicrobial-resistance profiling for the detected pathogens would substantially enhance the study's relevance to public health.

Response 5:

The aim of the research reported in this manuscript was to analyze the contamination differences and thereby risks associated with the ICLF and COF environments. Subsequent to the analyses reported here, the pathogens obtained from the samples were recently whole genome sequenced, and a manuscript will be prepared in the near future that will report and characterize the distribution of antibiotic resistance genes obtained from the farms.

Reviewer #2 (Comments for the Author):

Comment 1:

The manuscript is well written. However, the biggest issue is the stat analysis in the result section and the way they are presented in tables/figures, i.e. no stat indications of significance between sample groups. P values are only mentioned in the body of text, but are missing in all tables/figures.

Response 1:

Statistical indications of significance between sample groups have been added to the necessary tables/figures. Please see Figures (1, 2, & 3) and Table 6.

Re: Spectrum00862-25R1 (**Integrated Crop-Livestock Farming Systems Influence the Incidence of *Salmonella*, *Listeria monocytogenes*, Shiga toxin-producing *E. coli*, and Indicator Bacteria on Fresh Produce**)

Dear Dr. Salina Parveen:

Thank you for the privilege of reviewing your work. Below you will find my comments, instructions from the Spectrum editorial office, and the reviewer comments.

Revision Guidelines

Sincerely,
Dan Li
Editor
Microbiology Spectrum

Reviewer #1 (Comments for the Author):

The manuscript has several critical flaws that undermine its conclusions: it relies on a tiny, non-random sample of just three ICLF and two COF farms chosen via extension contacts, conflating "integrated vs. crop-only" with "organic vs. conventional" and leaving many confounders unaddressed; its sampling periods are mismatched-ICLFs from March 2020-December 2021, FMs from June-November 2021, and COFs/SMs from September-December 2021-so seasonal effects alone could explain apparent

differences; produce handling also varies (ICLF/COF samples were unwashed field harvests, whereas FM/SM produce may have been washed), and the types of crops differ (leafy greens vs. fruits), biasing contamination comparisons; statistically, all sample-level observations are treated as independent despite clustering within only a few farms, no mixed-effects models are used, and dozens of pairwise tests were run without adjusting for multiple comparisons, greatly increasing false-positive risk; many reported differences hinge on extremely low counts (e.g., Salmonella in 1/257 ICLF vs. 0/150 COF samples), yet no confidence intervals are provided and sparse data can yield misleading p-values; "control" soils that never received manure nonetheless showed very high pathogen levels (e.g., 55.6% Salmonella), indicating background contamination unrelated to ICLF practices; and finally, the authors overstate causality in an observational design by grouping ICLF with FM and COF with SM despite numerous uncontrolled differences in management, scale, and handling, making the findings neither robust nor generalizable. I strongly suggest adding WGS or AMR data to the manuscript.

Reviewer #3 (Comments for the Author):

Goodwyn et al. present results from their analysis of 1000+ samples from crop-only farms versus farms with integrated crop/animal production, with a convincing argument that that this type of study is important because foodborne outbreaks are increasing and the rising popularity of produce sold at farmers markets from integrated farms may be the culprit. The authors analyze many samples from a small subset of locations for several outcomes. I have several major comments:

1) The acronyms were very confusing. Suggest you do not abbreviate farmers market, supermarket, generic E. coli, L. monocytogenes, or total aerobic bacterial counts. Also do not abbreviate terms that are used fewer than 3 times, like NOP, PSR, FSMA, CEC, SOM

2) The authors consistently use "/" instead of just picking one term or using "and." Please reduce this as it is distracting. E.g. fruits/vegetables, guidelines/standards, isolation/confirmations, etc.

3) In the statistical analysis section, it is very challenging to follow why you did which analyses. Please clarify this section by explaining exactly what research question you are trying to answer each time you introduce a stats analysis. For example, "The relationship between pathogen/gEC prevalence and log coliform levels was also analyzed" - why? "For selected subsets of data considered separately, a logistic regression was fit for 300 each pathogen (and gEC).."- what is the first part of this sentence even referring to, and why was this analysis performed?

4) The study aims are listed in lines 111-117. Yet, the Results are presented in a different order than the aims, which makes the complex Results even more difficult to follow. For example, results related to soil health & fertility are presented first despite being the second aim of the study. Similarly, these results are discussed first despite not being the main focus of the study and not being set up/framed at all in the introduction. Suggest reordering.

5) Some results are presented by season, others are not. The distinction seems arbitrary at times and is challenging to follow.

6) The Discussion is extremely long and should be shortened. The authors should be clearer about what their study adds to the literature, instead of primarily describing results from other studies alongside their results.

7) The authors performed an enormous amount of statistical analyses, and as a previous reviewer pointed out, correcting for multiple hypothesis testing would make the authors' findings more robust.

Specific comments:

Line 78: Either eliminate "etc" or elaborate

Line 124: Was there anything different about farms or farmers markets that did not participate? How many farms & farmers markets are there in this setting and what fraction did you sample? How were supermarkets chosen?

Line 148: What is an "animal pen" sample? Soil, feces, surface swab, other?

Line 168-169: "Purchased in duplicate" - from the same supplier?

Line 182: Why measure APC and total coliforms?

Line 292 and line 295-295: What do you mean by "selected comparison categories"

Lines 306-307: The authors write "consistent with a per-comparison error rate control approach, p-values were unadjusted for the total number comparisons," presumably in response to a previous reviewer's comment. I have never seen this phrasing before. If I'm understanding right (and it's hard to know because there are no references to justify this approach), the authors are essentially saying: consistent with an approach where you do not take into account multiple comparisons, we did not take into account multiple comparisons." This needs far more justification and the sentence contains misspellings. I do not agree with this approach considering the authors performed well over 100 statistical tests.

Line 311 - what is CEC?

Line 313 - what is SOM?

Petrifilm Microbial Levels section - these numbers are very hard to parse. Suggest these results are visualized as boxplots rather than reported, or at least in addition to.

Line 374-376: What is the interpretation of this? What do these VFs represent? Same comment for lines 388-389.

Line 486: "Salmonella detections in our experiment"...what experimental work was done?

Line 494: "instead they may be below the analytical detection limit." Is your detection limit mentioned anywhere, and even if so, can you remind readers of it here?

Line 660-662: "and 66 were found to be statistically significant ($p < 0.05$) compared to 7 that would be expected by chance if the universal null hypothesis was true." How exactly was this determined?

Line 506: Again, the word "experiment" here is confusing

Table 1 - how did you decide what produce to focus on? Or were all types of produce selected?

Response to the Reviewers' comments

Thank you very much for taking your valuable time to read and give feedback to develop our manuscript. All comments have been addressed, and corresponding changes are highlighted in the marked changes version of the document. Also, line numbers in the responses indicate line numbers with changes or line numbers that support the responses.

Reviewer #1 (Comments for the Author):

The manuscript has several critical flaws that undermine its conclusions:

Comment 1: It relies on a tiny, non-random sample of just three ICLF and two COF farms chosen via extension contacts, conflating "integrated vs. crop-only" with "organic vs. conventional" and leaving many confounders unaddressed.

Response 1: In this study, a food safety extension specialist identified and connected the research team with at least 10 farms on the Maryland Eastern Shore that met the necessary criteria. All these farms were contacted, and the selected locations were those farms that 1) met the criteria listed in the manuscript, 2) could provide the necessary samples, and 3) agreed to participate in the study. In this study, there were sampled farms that previously worked with the extension program and farms that had not. Previously working with the university's extension program was not a determining factor in selecting locations. In a sense, farm locations were non-randomly selected; however, that's due to the nature of farms needing to meet certain criteria. Furthermore, since these were working farms that are separate from our university, locations could not be randomly selected; permission must be granted to access farms and obtain samples.

Though only a few farms participated, these farms reflect real-world practices of community-integrated crop-livestock farms (ICLFs) and crop-only farms (COFs). Here, the integrated crop-livestock farms were organic, and the crop-only farms were conventional. Yes, there are confounding variables unique to each location that may affect pathogen prevalence (in general, no two farms are exactly alike in all respects). Despite the lack of consistency, comparison studies, such as these, are necessary to observe, identify, and document the contamination differences and factors in real-world farm settings. Though this experiment could be performed in a controlled setting with consistent, definite variables, the results would not reflect actual agricultural practices. **To adjust for this, variables and how they affect results and pathogen prevalence are discussed.** For example, see lines 479 - 484, 508 - 511, and 555 - 560. Furthermore, Tables 2 and 3 illustrate the different farm management practices of each location.

Comment 2: Its sampling periods are mismatched-ICLFs from March 2020-December 2021, FMs from June-November 2021, and COFs/SMs from September-December 2021, so seasonal effects alone could explain apparent differences.

Response 2: This study began during the COVID-19 Pandemic, which limited farmer and sample availability. Pandemic restrictions affected the sampling timeline. Furthermore, samples were obtained from active, working farms, so samples and results reflect real-world farming conditions. However, this also means that sampling was dependent upon farmer availability and enrollment times. The differences in sampling periods and the number of samples collected were largely unavoidable as we were dependent upon farmer participation. Unfortunately, reweighting

to adjust for the resulting seasonal imbalance is problematic for several reasons and the only other alternative is to repeat our analyses on subsets of the data where the possibility of seasonal effects can be removed by subset selection. Since fall samples were collected from all four locations, we selected this season for a subset analysis and comparison to that obtained when analyses were applied to all data (see lines 310 – 312). At a minimum, where both analyses indicate similar differences in prevalence of pathogens, or abundance of APC/coliforms, the differences are less likely to be due to seasonal effects alone. For example, see lines 383-385, 395 - 397, and 630 - 634, where results of both analyses are presented and compared.

Comment 3: Produce handling also varies (ICLF/COF samples were unwashed field harvests, whereas FM/SM produce may have been washed), and the types of crops differ (leafy greens vs. fruits), biasing contamination comparisons.

Response 3: The main purpose of this study was to determine pathogen prevalence differences between traditional and non-traditional fresh produce production environments. The reason was to address the food safety concerns that arise from the differences in how these studied locations handle produce. Integrated crop livestock farms (ICLFs) apply manure-based amendments and raise animals in proximity to crop fields, which could increase contamination risks compared to crop-only farms (COFs). For this reason, ICLF soil and produce samples were compared to COF soil and produce. Produce sampled from farms was analyzed after harvesting from the soil, where no postharvest rinse steps were applied. **Farmers' markets are less stringently regulated than supermarkets.** For example, before products are sold at supermarkets, they are likely harvested, transported, and stored following standardized procedures (ex. refrigeration), whereas farmers' market foods may not incorporate these procedures in all cases. That is why farmers' market samples were compared to supermarket samples. See lines 118 – 121. These factors do bias contamination comparisons; however, it is necessary to make these comparisons to assess the food safety concerns in these real-world environments (lines 742-749).

The types of crops do slightly differ; however, **sample consistency was maintained in each environment to the best of our ability.** Four crop types (bell pepper, kale, squash, and tomato) were collected from each ICLF, COF, farmers' market, and supermarket locations. Furthermore, collard greens were collected from ICLF and COF locations as well. See Table 1. Since similar produce samples were collected from each sampling location, the study outcomes do provide insight into the contamination differences and risks associated with each location.

Comment 4: Statistically, all sample-level observations are treated as independent despite clustering within only a few farms, no mixed-effects models are used, and dozens of pairwise tests were run without adjusting for multiple comparisons, greatly increasing false-positive risk.

Response 4: Mixed effects models with farm as a random effect are not identifiable when applied to sparse count data with frequent counts of zero at the level of individual farms. And the fact that pathogen occurrence may vary over farms does not invalidate inferences about the frequency of occurrence in aggregated data, which is still meaningful even when farm-to-farm variability can't be well quantified. Modeling the clustering of pathogen occurrence over farms using mixed effects models also entails strong parametric assumptions that can be problematic and unverifiable when applied to sparse data. **Nonparametric methods such as Fisher's test do not require such assumptions.** As to false-positive risk associated with multiple tests, there is no imperative that demands adjustments for multiple comparisons when reporting the results of

research studies, as if it is a cut and dry issue with no pros and cons. In regard to the norms applicable to reporting the results of statistical analyses, the ASM instructions for authors refer to over a half dozen reporting guidelines developed for different types of human health research, including preclinical (observational) and clinical trial studies. With the possible exception of human clinical trials, none of these reporting guidelines are prescriptive about adjustments when reporting the results of multiple comparisons. **They emphasize comprehensive and transparent description, which can be achieved equally well irrespective of any adjustments for multiple comparisons.**

Comment 5: Many reported differences hinge on extremely low counts (e.g., Salmonella in 1/257 ICLF vs. 0/150 COF samples), yet no confidence intervals are provided, and sparse data can yield misleading p-values.

Response 5: There is no objective reason to believe that all other things being equal, p-values obtained using Fisher's test are less reliable (i.e., potentially misleading) when based on extremely low counts and small sample sizes. If the null hypothesis is true, the usual and customary Fisher's exact test with no mid-p correction is very conservative when applied to such data with an expected false positive rate that is less than the nominal significance level. This is a consequence of the test being exact and the discreteness of the potential outcomes. The test becomes less conservative when the counts are not extreme, and the total sample is large. Although confidence intervals for the odds ratio associated with Fisher's test can be provided, they are redundant in that they will exclude an odds ratio of 1 if and only if the p-value is less than the significance level and are, consequently, no more "reliable" than the corresponding p-values. Marginal confidence intervals on pathogen occurrence in the respective sample types are not of interest since they do not provide a confidence interval on differences in pathogen occurrence.

Comment 6: "Control" soils that never received manure nonetheless showed very high pathogen levels (e.g., 55.6% Salmonella), indicating background contamination unrelated to ICLF practices.

Response 6:

Control soils were located on the integrated crop livestock farms (ICLFs), as mentioned in the manuscript, a tracing analysis was not performed to determine the possible contamination vectors. In the discussion section, it is mentioned that background contamination could arise from domestic or wild animals and that pathogens could transfer as a result of farmers' activity or weather events (ex. wind, runoff, rain, etc.). See lines 632 – 634. In the current study and cited literature, contamination was present in ICLF and crop-only farm (COF) soils, indicating that practices specific to ICLFs are not the sole contamination risks in agricultural settings (lines 590 – 591 and 602 – 606).

Comment 7: And finally, the authors overstate causality in an observational design by grouping ICLF with FM and COF with SM despite numerous uncontrolled differences in management, scale, and handling, making the findings neither robust nor generalizable.

Response 7: In this manuscript, integrated crop livestock farms (ICLFs) with farmers' markets and crop-only farms (COFs) with supermarkets are not grouped. Instead, ICLFs are compared to COFs, and farmers' markets are compared to supermarkets. In a way, ICLFs and farmers'

markets are similar in that practices unique to these environments may leave produce at an increased risk for contamination. For example, the organic ICLFs in this study used manure-based amendments and raised animals in proximity to crop fields. Whereas the animals are not present on COFs, and these operations can apply synthetic fertilizers, pesticides, and antimicrobials. In terms of retail environments, farmers' markets are less stringently regulated compared to supermarkets, leaving farmers' market foods susceptible to increased contamination. In the discussion section, it is mentioned how the differences in management factors (ex. management, scale, handling, etc.) can affect pathogen prevalence. See lines 479 – 484, 508 – 511, 534 – 537, 555 – 560, 595 – 597, 652 – 659, 675 – 682. Between ICLF, COF, farmers' market, and supermarket environments, there are uncontrolled variables (management, scale, handling), which reflect real-world conditions. The results presented in this paper detail how these variables can influence pathogen prevalence in these environments. These results are generalizable in that the practices associated with each environment can help explain differences in contamination. **For example, ICLF soils, which received manure-based amendments, were more contaminated than COF soils, which did not receive manure-based amendments.**

Comment 8: I strongly suggest adding WGS or AMR data to the manuscript.

Response 8: We believe the information presented here contributes to the literature that describes food safety risks in pre- and postharvest fresh produce environments. Specifically, this manuscript details the contamination risks associated with ICLFs and farmers' markets and compares the risks to traditional crop-only farms and supermarkets, respectively. Adding WGS or AMR data to this manuscript would cause issues with the word limit. However, WGS is being performed on isolated *Salmonella*, *L. monocytogenes*, and STEC isolates from this project. Therefore, we are considering publishing WGS data in a separate manuscript.

Reviewer #3 (Comments for the Author):

Goodwyn et al. present results from their analysis of 1000+ samples from crop-only farms versus farms with integrated crop/animal production, with a convincing argument that this type of study is important because foodborne outbreaks are increasing and the rising popularity of produce sold at farmers markets from integrated farms may be the culprit. The authors analyze many samples from a small subset of locations for several outcomes. I have several major comments:

Comment 1: The acronyms were very confusing. Suggest you do not abbreviate farmers market, supermarket, generic *E. coli*, *L. monocytogenes*, or total aerobic bacterial counts. Also, do not abbreviate terms that are used fewer than 3 times, like NOP, PSR, FSMA, CEC, and SOM.

Response 1: The authors have revised and clarified the use of abbreviations throughout this manuscript.

Comment 2: The authors consistently use "/" instead of just picking one term or using "and." Please reduce this as it is distracting. E.g. fruits/vegetables, guidelines/standards, isolation/confirmations, etc.

Response 2: The authors have revised and reduced the use of combining terms with "/" throughout this manuscript.

Comment 3: In the statistical analysis section, it is very challenging to follow why you did which analyses. Please clarify this section by explaining exactly what research question you are trying to answer each time you introduce a stats analysis. For example, "The relationship between pathogen/gEC prevalence and log coliform levels was also analyzed" - why? "For selected subsets of data considered separately, a logistic regression was fit for 300 each pathogen (and gEC)." - what is the first part of this sentence even referring to, and why was this analysis performed?

Response 3: The statistical analysis section has been revised to clarify what comparison groups and what subsets of data were of interest for each of the four different types of analyses conducted. The first two analyses, concerning differences in abundance and prevalence, address a central research question of the paper, which is described in the introduction. Additional analyses were conducted to assess whether or not STEC serovars and VFs were associated with different groups of samples within each of several sample types and whether or not pathogen occurrence was related to abundance of coliforms. The latter analysis was conducted to determine a relationship between pathogen/generic *E. coli* prevalence and sanitary conditions (see lines 689 – 692 and 188 - 191).

Comment 4: The study aims are listed in lines 111-117. Yet, the Results are presented in a different order than the aims, which makes the complex Results even more difficult to follow. For example, results related to soil health & fertility are presented first despite being the second aim of the study. Similarly, these results are discussed first despite not

being the main focus of the study and not being set up/framed at all in the introduction. Suggest reordering.

Response 4: The authors have reordered the manuscript sections to follow the order of the study aims. The use of manure-based amendments to improve soil health and fertility is briefly mentioned in the introduction. However, it cannot be expounded upon due to word limitations. Here, soil health was not the main goal of the study. It was analyzed to see the potential of these amendments as soil fertilizers in community, integrated crop-livestock farm environments.

Comment 5: Some results are presented by season, others are not. The distinction seems arbitrary at times and is challenging to follow.

Response 5: The seasonal results were not presented arbitrarily. In this manuscript seasonal results were presented for two reasons. First, seasonal results were presented to describe when most (or the sole) detections occurred. For example, see lines 364 – 367 and 371 – 373. This provides insight into when pathogen incidence may be at an increased risk. Second, analyses of the fall subset of data were conducted because the collected sample data is seasonally imbalanced and that potentially affects the results of analyses using all of the data. For example, all crop-only farm (COF) samples were collected in the fall season and integrated crop livestock farm (ICLF) samples were collected over multiple seasons, including the fall. Comparisons between ICLF and COF based on the fall subset of data are unaffected by the seasonal imbalance and are of interest for that reason in comparison to the results of analyses applied to all of the data. For example, see lines 383 – 385, 395 – 397, and 625 - 628. Both these presentations of seasonal results in this manuscript are needed to provide insight into contamination risks and provide a more comprehensive comparison between ICLF and COF samples.

Comment 6: The Discussion is extremely long and should be shortened. The authors should be clearer about what their study adds to the literature, instead of primarily describing results from other studies alongside their results.

Response 6: The description of results from other studies has been reduced. Statements of what the study adds to the literature have been made. Specifically, the study analyzed contamination risks in real-world environments that reflect real-world practices (lines 473 - 475). Contributions to literature can be derived from the discussed variables that select for pathogens and how these need to be addressed to improve the safety of local ICLFS and farmers' markets.

- How pathogen prevalence is affected by multiple animal species (Lines 479 – 484)
- Manure management strategies (lines 508 - 511).
- Farm activities such as U-pick (Lines 595 - 597).
- Farmers market food storage and display (lines 675 - 682).

Comment 7: The authors performed an enormous amount of statistical analyses, and as a previous reviewer pointed out, correcting for multiple hypothesis testing would make the authors' findings more robust.

Response 7: One can argue that the significance level of any hypothesis test in any research paper should be set lower than 0.05 and the smaller that significance level the more “robust” the declaration of significance will be regardless of whether it's the only hypothesis test in a research

paper or one of a 100. But this is only “robust” in the sense of controlling type I error and it comes at the expense of less statistical power (and more type II error), which is no less important and consequential. That is not a good trade off in all circumstances and unadjusted p-values are very interpretable, particularly if you know how many false positives to expect if the universal null hypothesis is true. There are many research contexts where this is very common. The amount of statistical analysis in the paper is hardly enormous.

Comment 8: Line 78: Either eliminate "etc." or elaborate.

Response 8: The authors have eliminated etc. and revised the sentence. See lines 77 – 80.

Comment 9: Line 124: Was there anything different about farms or farmers’ markets that did not participate? How many farms & farmers’ markets are there in this setting and what fraction did you sample? How were supermarkets chosen?

Response 9: The criteria for how farms and farmers’ markets were selected is detailed in the manuscript (See lines 126 – 138). Farms that were not selected either did not respond to our invitation or did not have the needed samples available. Agriculture is a staple on the Eastern Shore of Maryland, with numerous backyard, small-scale, and large-scale producers that may or may not be registered with local directories. Therefore, it is difficult to say how many farms are present. Regarding farmers’ markets, there are 14 on the Eastern Shore of Maryland that are registered with the Maryland Farmers’ market directory. The 3 integrated crop livestock farms (ICLF), 2 crop-only farms (COF), 3 Farmers’ markets, and 3 supermarkets were selected because they had the needed samples and met the criteria.

Comment 10: Line 148: What is an "animal pen" sample? Soil, feces, surface swab, other?

Response 10: Animal pen samples consisted of soils collected at a depth of ~7 cm from high-trafficked areas (water/feed) of animal enclosures where manures were collected, stored, and aged for an ICLF’s manure amendment. This is detailed in the sample collection section. See lines 171 – 173.

Comment 11: Line 168-169: "Purchased in duplicate" - from the same supplier?

Response 11: At the time of sampling, produce samples were purchased in duplicate from each vendor or supplier at farmers’ markets and supermarkets. This is detailed in the manuscript (lines 179 - 181).

Comment 12: Line 182: Why measure APC and total coliforms?

Response 12: Total APC and coliforms were measured to assess the general microbiological status conditions associated with samples obtained from ICLF, COF, farmers’ market, and supermarket locations. See lines 188 – 191.

Comment 13: Line 292 and line 295-295: What do you mean by "selected comparison categories"

Response 13: The phrase “selected comparison categories” was a general term to briefly describe how samples were grouped for comparisons through statistical analysis. However, the

statistical analysis section has been revised. Instead of this phrase, comparison groups have been mentioned throughout the manuscript. For example, the introduction mentions that samples from ICLFs were compared to corresponding samples from crop-only farms, and samples from farmers' markets were compared to corresponding samples from supermarkets (lines 118 – 121). Also, the comparisons of interests and how samples were grouped based on sample type are described in the statistical analysis section (lines 306 - 310). Lines 315 - 319 even explains how samples were grouped and compared for ANOVA analysis.

Comment 14: Lines 306-307: The authors write "consistent with a per-comparison error rate control approach, p-values where unadjusted for the total number comparisons," presumably in response to a previous reviewer's comment. I have never seen this phrasing before. If I'm understanding right (and it's hard to know because there are no references to justify this approach), the authors are essentially saying: consistent with an approach where you do not take into account multiple comparisons, we did not take into account multiple comparisons." This needs far more justification and the sentence contains misspellings. I do not agree with this approach considering the authors performed well over 100 statistical tests.

Response 14: The phrase “consistent with a per-comparison error rate control approach” in the middle of that sentence is simply putting a label on the stated approach. The phrasing is consistent with that used in many statistical textbooks and research articles on the topic of multiple comparisons (“per-comparison error” is often abbreviated as PCE and it is common enough to have a Wikipedia page). And it should be clear enough from the fact that that sentence is in the methods section that it is a declarative statement as to what method was selected and applied and not a justification as to why it was selected. Discussion as to why a particular method was selected is usually part of the discussion if it merits discussion at all. We’ve added a reference to the discussion section but, again, the perception that adjustments for multiple comparisons are obligatory is misinformed and not consistent with what is required by most reporting guidelines (see response 4 to reviewer 1).

Comment 15: Line 311 - what is CEC?

Response 15: This abbreviation is cation exchange capacity, which is the soil’s ability to retain positive charged ions such as K^+ , Ca^+ , Mg^+ . This measurement provides insights into soils nutrient retention capacity. This abbreviation has been removed, and the term has been explained in the article. See lines 462 - 464.

Comment 16: Line 313 - what is SOM?

Response 16: This abbreviation is soil organic matter, which is decomposed, organic materials (plant, animal, soil microbes) in the soil. Soil organic matter benefits soils improving soil structure, nutrient cycling, and water retention, and reduces erosion. This abbreviation has been removed, and the term has been explained in the article. See lines 465 – 467.

Comment 17: Petrifilm Microbial Levels section - these numbers are very hard to parse. Suggest these results are visualized as boxplots rather than reported, or at least in addition to.

Response 17: Box plots of the Petrifilm levels across the various sampling locations and types have been created and included in the manuscript as figure 1.

Comment 18: Line 374-376: What is the interpretation of this? What do these VFs represent? Same comment for lines 388-389.

Response 18: Virulence factors (VFs) contribute to pathogenicity. The prevalence of these VFs indicates that pathogen genes related to severe cases and foodborne outbreaks associated with STEC are prevalent in tested samples. This information provides insight into food safety risks associated with each sample type and environment. The reasoning behind choosing these VFs has been added. See lines 276 - 278.

Comment 19: Line 486: "Salmonella detections in our experiment" ...what experimental work was done? (Line 468)

Response 19: Here, the authors were describing from what types of animals the BSAAO *Salmonella* detections occurred. In this case, the *Salmonella* detections originated from BSAAOs containing chicken manure. The word experiment was to point to the fact that results from the current study were being discussed. This sentence has been clarified. The experimental work is detailed in the Materials and Methods section of this manuscript. Lines 513 - 514.

Comment 20: Line 494: "instead they may be below the analytical detection limit." Is your detection limit mentioned anywhere, and even if so, can you remind readers of it here?

Response 20: The product description describes a detection limit of one organism per 100 mL. This detail has been added to the manuscript. Lines 205 - 207, 520 - 522.

Comment 21: Line 660-662: "and 66 were found to be statistically significant ($p < 0.05$) compared to 7 that would be expected by chance if the universal null hypothesis was true." How exactly was this determined?

Response 21: If the universal null hypothesis is true then the p-values obtained are a sample of 130 independent uniform random variables each of which has a probability of 0.05 of being called significant ($p < 0.05$). Adding up these 130 probabilities (130×0.05) gives 6.5 which we rounded up to 7.

Comment 22: Line 506: Again, the word "experiment" here is confusing

Response 22: Here, the authors were describing from where pathogens were isolated in this study. In this case, pathogens were isolated from ICLF-produce harvested from soils that received BSAAO amendments. The word experiment was to point to the fact that results from the current study were being discussed. This sentence has been clarified. The experimental work is detailed in the Materials and Methods section of this manuscript. Lines 534 - 535.

Comment 23: Table 1 - how did you decide what produce to focus on? Or were all types of produce selected?

Response 23: The goal was to focus on produce that is eaten raw or grown in close contact with the soil. Selected Produce types also had to be readily available at multiple locations. This has been indicated in the Table 1 footnote.

Re: Spectrum00862-25R2 (**Integrated Crop-Livestock Farming Systems Influence the Incidence of *Salmonella*, *Listeria monocytogenes*, Shiga toxin-producing *E. coli*, and Indicator Bacteria on Fresh Produce**)

Dear Dr. Salina Parveen:

Your manuscript has been accepted, and I am forwarding it to the ASM production staff for publication. Your paper will first be checked to make sure all elements meet the technical requirements. ASM staff will contact you if anything needs to be revised before copyediting and production can begin. Otherwise, you will be notified when your proofs are ready to be viewed.

Sincerely,
Dan Li
Editor
Microbiology Spectrum

Reviewer #1 (Comments for the Author):

The authors have addressed all the comments